# SCALABLE MECHANISTIC NEURAL NETWORKS

**Jiale Chen, Dingling Yao, Adeel Pervez, Dan Alistarh, Francesco Locatello**
Institute of Science and Technology Austria (ISTA)
`jiale.chen@ist.ac.at`

## ABSTRACT

We propose *Scalable* Mechanistic Neural Network (S-MNN), an enhanced neural network framework designed for scientific machine learning applications involving long temporal sequences. By reformulating the original Mechanistic Neural Network (MNN) (Pervez et al., 2024), we reduce the computational time and space complexities from cubic and quadratic with respect to the sequence length, respectively, to *linear*. This significant improvement enables efficient modeling of long-term dynamics without sacrificing accuracy or interpretability. Extensive experiments demonstrate that S-MNN matches the original MNN in precision while substantially reducing computational resources. Consequently, S-MNN can drop-in replace the original MNN in applications, providing a practical and efficient tool for integrating mechanistic bottlenecks into neural network models of complex dynamical systems. Source code is available at `https://github.com/IST-DASLab/ScalableMNN`.

## 1 INTRODUCTION

The Mechanistic Neural Network (MNN) (Pervez et al., 2024) has recently emerged as a promising approach in scientific machine learning. Unlike traditional black-box approaches for dynamical systems (Chen et al., 2018; 2021; Kidger et al., 2021; Norcliffe et al., 2020) that primarily focus on forecasting, MNN additionally learns an explicit internal ordinary differential equation (ODE) representation from the noisy observational data that enables various downstream scientific analysis such as parameter identification and causal effect estimation (Yao et al., 2024). Despite their advantages, the original formulation of MNN faces significant scalability challenges. Specifically, the computational time and memory usage severely limit the practical applicability of MNN to problems involving long time horizons or high-resolution temporal data, such as climate recordings (Verma et al., 2024), as the required computations become prohibitive even for the most advanced hardware.

The inefficiency stems from the matrix operations required to solve the linear systems associated with the MNN. In the original framework, two solvers are provided: a *dense* solver and a *sparse* solver. The *dense* solver operates on dense matrices and employs standard methods for solving linear systems, resulting in cubic time and quadratic space complexities with respect to the sequence length. This computational inefficiency makes it unsuitable for long sequences. The MNN *sparse* solver, on the other hand, constructs sparse matrices and uses iterative methods such as the conjugate gradient algorithm to solve the linear systems. While this reduces memory usage by exploiting sparsity, the unstructured sparsity patterns of the matrices prevent the

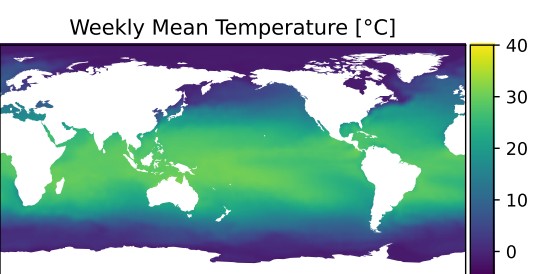

Figure 1: 4-year sea surface temperature (SST) forecasting using the Mechanistic Identifier (Yao et al., 2024) and our *Scalable* Mechanistic Neural Network (S-MNN).

solver from fully leveraging the GPU's parallelism potential. Additionally, iterative methods can suffer from slow convergence and numerical inaccuracies, particularly for large-scale problems.

This work proposes a *scalable* variant of MNN (S-MNN) that reduces the computational time and space complexities from cubic and quadratic with respect to the sequence length, respectively, to *linear*, while maintaining on-par accuracy. As a result, we successfully demonstrate the real-world applicability (Figure 1) of S-MNN on long-term climate data that the original MNN failed to handle. Our main contributions are as follows:

- **Complexity Reduction.** We reformulate the original MNN's underlying linear system by eliminating the slack variables and central difference constraints, and reducing the quadratic programming problem to least squares regression (Section 3.1). This results in the left-hand side square matrix having a banded structure, allowing us to employ efficient algorithms (Section 3.2). The time and space complexities are reduced to *linear* with respect to the sequence length, making it suitable for long-sequence modeling.

- **Efficient Solver Design.** We develop an efficient solver that leverages the inherent sparsity and banded structure of the reformulated linear system (Section 3.2). The solver is optimized for GPU execution, fully exploiting parallelism to achieve significant speed-ups.

- **Long-Term Sequence Modeling for Science.** We validate the effectiveness of our S-MNN through experiments on various benchmarks, including governing equation discovery with the Lorenz system (Section 5.2), solving the Korteweg-de Vries (KdV) partial derivative equation (Section 5.3), and sea surface temperature (SST) prediction for modeling long real-world temporal sequences (Section 5.4). Our results demonstrate that S-MNN matches the precision of the original MNN while significantly reducing computational time and memory usage across the board.

## 2 OVERVIEW OF MECHANISTIC NEURAL NETWORKS

The Mechanistic Neural Network (MNN) by Pervez et al. (2024) consists of three components: a mechanistic encoder, a specialized differentiable ordinary differential equation (ODE) solver based on constrained optimization, and a mechanistic decoder. The mechanistic encoder, realized as a neural network, generates a semi-symbolic representation of the underlying ODE from an input time series, effectively learning the dynamics from the data. The solver then constructs and solves a linear system that equivalently describes the original system. The mechanistic decoder processes the solver solutions to the final outputs.

For tasks such as forecasting future sequences from past data, the encoder processes the input data to produce the trajectory-specific semi-symbolic representation. Additionally, the encoder can be designed to overcome the limitations of the linear ODE solver by learning parameters of nonlinear basis functions (Brunton et al., 2016a), enabling the MNN to model nonlinear ODEs effectively. The decoder is optional and their necessity depends on the specific task used for training.

Formally, given a multidimensional discretized time sequence $\boldsymbol{x}_1, \boldsymbol{x}_2, \ldots, \boldsymbol{x}_T$ as the input, the mechanistic encoder maps $\boldsymbol{x}_{1:T}$ into semi-symbolic representations of a set of linear ODEs

$$\sum_{r=0}^{R} \boldsymbol{c}_r^\top \left(t, \boldsymbol{x}_{1:T}\right) \frac{\mathrm{d}^r \boldsymbol{y}}{\mathrm{d}t^r} = d\left(t, \boldsymbol{x}_{1:T}\right) \tag{1}$$

of the time $t$ dependent variable $\boldsymbol{y} \in \mathbb{R}^V$ with the initial conditions

$$\frac{\mathrm{d}^r \boldsymbol{y}}{\mathrm{d}t^r} = \boldsymbol{u}_r\left(\boldsymbol{x}_{1:T}\right) \tag{2}$$

where $V$ is the dimension of $\boldsymbol{y}$, $R$ is the highest derivative order, $\boldsymbol{c}_r\left(t, \boldsymbol{x}_{1:T}\right) \in \mathbb{R}^V$ and $d\left(t, \boldsymbol{x}_{1:T}\right) \in \mathbb{R}$ are the coefficients, and $\boldsymbol{u}_r\left(\boldsymbol{x}_{1:T}\right) \in \mathbb{R}^V$ represents the initial conditions. $\boldsymbol{c}_r\left(t, \boldsymbol{x}_{1:T}\right)$, $d\left(t, \boldsymbol{x}_{1:T}\right)$, $\boldsymbol{u}_r\left(\boldsymbol{x}_{1:T}\right)$, as well as the step sizes $s\left(\boldsymbol{x}_{1:T}\right)$ for the time discretization are learned by the encoder. The representation is compactly denoted as $\{c, d, u, s\}$. The MNN solver then solves the linear ODE using this representation, producing the discretized output time sequence $\boldsymbol{y}_1, \boldsymbol{y}_2, \ldots, \boldsymbol{y}_T$. The decoder takes $\boldsymbol{y}_{1:T}$ as input and generates the final output sequence $\boldsymbol{z}_1, \boldsymbol{z}_2, \ldots, \boldsymbol{z}_T$. A loss $\ell$ can be computed based on $\boldsymbol{z}_{1:T}$ and the target data, and the encoder and decoder networks are updated using gradient descent methods.

An illustrative example is the task of discovering a $V$-dimensional coefficient $\boldsymbol{\xi} \in \mathbb{R}^V$ for a time-independent one-dimensional first-order ODE in the form of

$$\mathrm{d}y/\mathrm{d}t = g\left(\boldsymbol{\xi}^\top \phi\left(y\right)\right) \tag{3}$$

where $\phi : \mathbb{R} \to \mathbb{R}^V$ is a $V$-dimensional non-linear basis function, $g : \mathbb{R} \to \mathbb{R}$ is a differentiable function. In this task, we first generate an initial $\boldsymbol{\xi}$ and specify the initial condition $u_{r=0} = x_1$. We fix $c$ to ones where they are multiplied with the first-order derivatives, and zeros otherwise. The step sizes $s$ are determined by the dataset's time increments. During each gradient descent iteration, we set $d = g\left(\boldsymbol{\xi}^\top \phi(x)\right)$, and then the MNN solver solves for a discretized solution $y_{1:T}$ for Eq. 3. We update $\boldsymbol{\xi}$ by minimizing the loss $\ell = \sum_{i=1}^{T} \|y_i - x_i\|^2$.

# 3 SCALABLE MECHANISTIC NEURAL NETWORKS

In this section, we define the linear system for the *Scalable* Mechanistic Neural Network (S-MNN) in Subsection 3.1, and present the solver's implementation and complexity analysis in Subsection 3.2.

## 3.1 LINEAR SYSTEM FORMULATION

A linear ordinary differential equation (ODE) system can be characterized by a set of linear equations involving the successive derivatives of an unknown time-dependent function $y$. Formally, in a system with $V$ variables (output dimensions), derivative orders up to $R$, and $Q$ governing equations, the state at $T$ discrete time points can be described by a series of clauses

$$\sum_{v=1}^{V} \sum_{r=0}^{R} c_{t,q,v,r} y_{t,v,r} = d_{t,q}, \quad \forall t \in \{1, \dots, T\}, q \in \{1, \dots, Q\}, \tag{4}$$

where $y_{t,v,r}$ is the $r$-th derivative of the $v$-th variable at $t$-th time point, $c_{t,q,v,r}$ is the corresponding coefficient for the $q$-th governing equation, and $d_{t,q}$ is a constant term.

Initial values $u_{t,v,r}$ are specified for each $y_{t,v,r}$ up to time point $T_{\text{init}}$ $(1 \leq T_{\text{init}} \leq T)$ and derivative order $R_{\text{init}}$ $(0 \leq R_{\text{init}} \leq R)$:

$$y_{t,v,r} = u_{t,v,r}, \quad \forall t \in \{1, \dots, T_{\text{init}}\}, v \in \{1, \dots, V\}, r \in \{0, \dots, R_{\text{init}}\}. \tag{5}$$

To ensure the smoothness of the trajectory, i.e., that the computed higher-order derivatives are consistent with the derivatives of lower-order terms, we introduce smoothness constraints. We define the forward and backward smoothness constraints using the Taylor expansions of the function $y$ at time points $t$ and $t+1$ respectively:

$$y_{t+1,v,r} = \sum_{r'=r}^{R} \frac{s_t^{r'-r}}{(r'-r)!} y_{t,v,r'}, \quad \forall t \in \{1, \dots, T-1\}, v \in \{1, \dots, V\}, r \in \{0, \dots, R\}, \tag{6}$$

$$y_{t,v,r} = \sum_{r'=r}^{R} \frac{(-s_t)^{r'-r}}{(r'-r)!} y_{t+1,v,r'}, \quad \forall t \in \{1, \dots, T-1\}, v \in \{1, \dots, V\}, r \in \{0, \dots, R\}. \tag{7}$$

where $s_t$ is the time span between time points $t$ and $t+1$.

Combining the constraints Eqs. 4, 5, 6, and 7 yields a linear system. In total there are $m$ constraints and $n$ unknown variables where

$$m = TQ + T_{\text{init}}V(R_{\text{init}}+1) + 2(T-1)V(R+1) \quad \text{and} \quad n = TV(R+1). \tag{8}$$

We arrange the unknown variables $y_{t,v,r}$ into a vector $\boldsymbol{y} \in \mathbb{R}^n$, the left-hand side coefficients into a matrix $\boldsymbol{A} \in \mathbb{R}^{m \times n}$, and the right-hand side into a vector $\boldsymbol{b} \in \mathbb{R}^m$. The linear system can then be compactly represented as $\boldsymbol{A}\boldsymbol{y} = \boldsymbol{b}$.

This system is over-determined $(m > n)$ under typical conditions $(T > 1)$ and can be solved for $\boldsymbol{y}$ using least squares regression. To balance the contributions of different constraints, we weight the smoothness constraints (Eqs. 6 and 7) by $s_t^r$. Additionally, we introduce optional importance weights $w_{\text{gov}}, w_{\text{init}}, w_{\text{smooth}} \in \mathbb{R}$, applied to the governing equations (Eq. 4), initial conditions (Eq. 5), and smoothness constraints (Eqs. 6 and 7), respectively. This flexibility allows for emphasizing specific aspects of the model during optimization. These weights are encoded into a diagonal matrix $\boldsymbol{W}$, and the solution for $\boldsymbol{y}$ is then given by:

$$\boldsymbol{y}(c, d, u, s) = \left(\boldsymbol{A}^\top \boldsymbol{W} \boldsymbol{A}\right)^{-1} \boldsymbol{A}^\top \boldsymbol{W} \boldsymbol{b}. \tag{9}$$

Note that $\boldsymbol{y}$ is differentiable with respect to $c$, $d$, $u$, and $s$. We provide more details on the formulations of $\boldsymbol{A}$, $\boldsymbol{b}$, $\boldsymbol{W}$, and $\boldsymbol{y}$ in Appendix A.1.

Our S-MNN formulation has three key differences from the original MNN formulation in Pervez et al. (2024): (1) the slack variables in the smoothness constraints are removed; (2) the forward and backward smoothness constraints are extended to the highest order ($r = R$), replacing the central difference constraints; (3) the quadratic programming is replaced by a least squares regression. For a supplementary description of the aforementioned MNN aspects, please refer to Appendix C.

Also note that, unlike the finite difference method which approximates derivatives by discretizing differential equations, our approach formulates a linear system to directly involve the derivative terms.

### 3.2 SOLVER DESIGN, ARCHITECTURE, AND ANALYSIS

The primary motivation for our improvement is our observation that, if we remove the slack variables, the matrix $\boldsymbol{A}$ will exhibit a specific sparsity pattern that can be exploited for computational and memory gains. A direct implementation based on the dense matrix $\boldsymbol{A}$ using Eq. 9 incurs cubic time complexity and quadratic space complexity due to matrix multiplication and inversion. However, by analyzing the sparsity pattern of $\boldsymbol{A}$, we find that $1 - O(1/T)$ of the values in the intermediate steps are zero and do not need to be computed or stored. In more detail, Eq. 9 can be decomposed into two steps: (1) a matrix-matrix multiplication $\boldsymbol{M} = \left(\boldsymbol{A}^\top \boldsymbol{W}\right)\boldsymbol{A}$ and a matrix-vector multiplication $\boldsymbol{\beta} = \left(\boldsymbol{A}^\top \boldsymbol{W}\right)\boldsymbol{b}$; (2) solving for $\boldsymbol{y}$ via the linear system $\boldsymbol{M}\boldsymbol{y} = \boldsymbol{\beta}$. **The key idea is to structure $\boldsymbol{M}$ as a banded matrix.** Observing the constraints, we note that the coefficients at time point $t$ are directly related only to those at time points $t-1$, $t$, and $t+1$. By ordering the variables by $t$, we can arrange $\boldsymbol{M}$ into a banded matrix. Specifically, the variable $y_{t,v,r}$ is placed at the position $((t-1)V + v - 1)(R+1) + r + 1$ in the vector $\boldsymbol{y}$, and the columns of matrix $\boldsymbol{A}$ are ordered accordingly. The resulting $\boldsymbol{M}$ is a symmetric matrix in a block-banded form:

$$\boldsymbol{M} = \begin{bmatrix} \boldsymbol{M}_1 & \boldsymbol{N}_1^\top & & \\ \boldsymbol{N}_1 & \boldsymbol{M}_2 & \ddots & \\ & \ddots & \ddots & \boldsymbol{N}_{T-1}^\top \\ & & \boldsymbol{N}_{T-1} & \boldsymbol{M}_T \end{bmatrix} \tag{10}$$

where each block is a square matrix of size $V(R+1)$. It is important to note that such a banded form is only possible after removing the slack variables from the original MNN formulation, as the slack variables introduce direct relationships between components at all time points.

For the matrix $\boldsymbol{M}$, only the non-zero blocks $\boldsymbol{M}_t$ and $\boldsymbol{N}_t$ need to be computed and stored. This can be achieved using efficient dense matrix operations with appropriately formatted dependencies $c$, $d$, $u$, and $s$. The matrix $\boldsymbol{A}$ does not need to be explicitly constructed. We present the detailed forward pass calculations for computing $\boldsymbol{M}$ and $\boldsymbol{\beta}$ in Appendix A.2 and omit them in this subsection. The backward pass is supported by automatic differentiation.

For solving the linear system $\boldsymbol{M}\boldsymbol{y} = \boldsymbol{\beta}$, we propose an efficient GPU-friendly algorithm (Algorithm 1). Specifically, because $\boldsymbol{M} = \boldsymbol{A}^\top \boldsymbol{W}\boldsymbol{A}$, $\boldsymbol{M}$ is positive-definite, and we can factorize $\boldsymbol{M}$ to a banded lower triangular matrix $\boldsymbol{P}$ and a block diagonal lower triangular matrix $\boldsymbol{L}$ using blocked versions of LDL and Cholesky decompositions such that

$$\boldsymbol{P}\boldsymbol{L}\boldsymbol{L}^\top \boldsymbol{P}^\top = \boldsymbol{M} \tag{11}$$

where $\boldsymbol{P}$ and $\boldsymbol{L}$ are partitioned into block matrices in the same form as $\boldsymbol{M}$:

$$\boldsymbol{P} = \begin{bmatrix} \boldsymbol{I} & & & \\ \boldsymbol{P}_1 & \boldsymbol{I} & & \\ & \ddots & \ddots & \\ & & \boldsymbol{P}_{T-1} & \boldsymbol{I} \end{bmatrix} \quad \text{and} \quad \boldsymbol{L} = \begin{bmatrix} \boldsymbol{L}_1 & & & \\ & \boldsymbol{L}_2 & & \\ & & \ddots & \\ & & & \boldsymbol{L}_T \end{bmatrix}. \tag{12}$$

with each block as a square matrix of size $V(R+1)$ and the diagonal blocks of $\boldsymbol{P}$ as an identity matrix $\boldsymbol{I}$. We also partition $\boldsymbol{\beta}$ and $\boldsymbol{y}$ into sub-vectors, $\boldsymbol{\beta} = \left[\boldsymbol{\beta}_1^\top, \boldsymbol{\beta}_2^\top, \ldots, \boldsymbol{\beta}_T^\top\right]^\top$ and

$\boldsymbol{y} = \left[ \boldsymbol{y}_1^\top, \boldsymbol{y}_2^\top, \ldots, \boldsymbol{y}_T^\top \right]^\top$. We then use forward and backward substitution to solve the following sequence of equations:

$$\boldsymbol{P}\boldsymbol{\beta}' = \boldsymbol{\beta}, \quad \boldsymbol{L}\boldsymbol{\beta}'' = \boldsymbol{\beta}', \quad \boldsymbol{L}^\top \boldsymbol{\beta}''' = \boldsymbol{\beta}'', \quad \boldsymbol{P}^\top \boldsymbol{y} = \boldsymbol{\beta}'''. \tag{13}$$

The main advantage of this algorithm is that only the blocks $\boldsymbol{P}_t$ and $\boldsymbol{L}_t$ in the matrices $\boldsymbol{P}$ and $\boldsymbol{L}$ need to be computed and stored, which limits the computational complexities of the LDL and Cholesky decompositions and the subsequent forward and backward substitutions to be linear in $T$. In contrast, explicitly computing $\boldsymbol{M}^{-1}$ or $\boldsymbol{L}^{-1}$ would involve dense or triangular-dense full-size matrices and should be avoided.

Interestingly, for the backward pass of $\boldsymbol{M}\boldsymbol{y} = \boldsymbol{\beta}$, the back-propagated gradients of $\boldsymbol{M}$ and $\boldsymbol{\beta}$ have elegant analytic solutions that enable further (constant factor) speed-ups compared to automatic differentiation. Assuming that the final loss is $\ell$, and given $\partial\ell/\partial\boldsymbol{y}$, the gradients are

$$\frac{\partial\ell}{\partial\boldsymbol{\beta}} = \boldsymbol{M}^{-1}\frac{\partial\ell}{\partial\boldsymbol{y}} \quad \text{and} \quad \frac{\partial\ell}{\partial\boldsymbol{M}} = -\frac{\partial\ell}{\partial\boldsymbol{\beta}}\boldsymbol{y}^\top. \tag{14}$$

The proof can be found in Appendix A.3. Algorithm 2 details the backward pass. Computing $\partial\ell/\partial\boldsymbol{\beta}$ involves solving a similar linear system using the banded matrix $\boldsymbol{M}$, and $\partial\ell/\partial\boldsymbol{M}$ is a vector outer product that can be efficiently computed for the non-zero blocks $\boldsymbol{M}_t$ and $\boldsymbol{N}_t$. The matrices $\boldsymbol{P}$, $\boldsymbol{L}$, and the solution $\boldsymbol{y}$ can be stored during the forward pass and reused for the backward pass.

In Appendix A.4, we present the training (Algorithm 5) and testing (Algorithm 6) procedures for a generic application of our S-MNN framework to help readers have a better overview.

---

**Algorithm 1:** Solver Forward Pass

**Input:** $\boldsymbol{M}_{1:T}, \boldsymbol{N}_{1:T-1}, \boldsymbol{\beta}_{1:T}$
**Output:** $\boldsymbol{L}_{1:T}, \boldsymbol{P}_{1:T-1}, \boldsymbol{y}_{1:T}$
1   $\boldsymbol{L}_{1:T}, \boldsymbol{P}_{1:T-1}$
     $\leftarrow \text{DECOMPOSE}\left(\boldsymbol{M}_{1:T}, \boldsymbol{N}_{1:T-1}\right)$;
2   $\boldsymbol{y}_{1:T} \leftarrow \text{SUBSTITUTE}\left(\boldsymbol{L}_{1:T}, \boldsymbol{P}_{1:T-1}, \boldsymbol{\beta}_{1:T}\right)$;

---

**Algorithm 3:** Decompose

**Input:** $\boldsymbol{M}_{1:T}, \boldsymbol{N}_{1:T-1}$
**Output:** $\boldsymbol{L}_{1:T}, \boldsymbol{P}_{1:T-1}$
1   $\boldsymbol{L}_{1:T}, \boldsymbol{P}_{1:T-1} \leftarrow \boldsymbol{M}_{1:T}, \boldsymbol{N}_{1:T-1}$;
2   **for** $t \leftarrow 1$ to $T$ **do**
     /* blockwise Cholesky */
3     **if** $t > 1$ **then**
4       $\boldsymbol{P}_{t-1} \leftarrow \boldsymbol{P}_{t-1}\boldsymbol{L}_{t-1}^{-\top}$;
5       $\boldsymbol{L}_t \leftarrow \boldsymbol{L}_t - \boldsymbol{P}_{t-1}\boldsymbol{P}_{t-1}^\top$;
6     **end**
7     $\boldsymbol{L}_t \leftarrow \text{CHOLESKY}\left(\boldsymbol{L}_t\right)$;
     // standard Cholesky
8   **end**
9   **for** $t \leftarrow 1$ to $T-1$ *in parallel* **do**
     /* to blockwise LDL */
10   $\boldsymbol{P}_t \leftarrow \boldsymbol{P}_t\boldsymbol{L}_t^{-1}$;
11 **end**

---

**Algorithm 2:** Solver Backward Pass

**Input:** $\boldsymbol{L}_{1:T}, \boldsymbol{P}_{1:T-1}, \boldsymbol{y}_{1:T}, \frac{\partial\ell}{\partial\boldsymbol{y}_{1:T}}$
**Output:** $\frac{\partial\ell}{\partial\boldsymbol{M}_{1:T}}, \frac{\partial\ell}{\partial\boldsymbol{N}_{1:T-1}}, \frac{\partial\ell}{\partial\boldsymbol{\beta}_{1:T}}$
1   $\frac{\partial\ell}{\partial\boldsymbol{\beta}_{1:T}}$
     $\leftarrow \text{SUBSTITUTE}\left(\boldsymbol{L}_{1:T}, \boldsymbol{P}_{1:T-1}, \frac{\partial\ell}{\partial\boldsymbol{y}_{1:T}}\right)$;
2   **for** $t \leftarrow 1$ to $T$ *in parallel* **do**
3     $\frac{\partial\ell}{\partial\boldsymbol{M}_t} \leftarrow -\frac{\partial\ell}{\partial\boldsymbol{\beta}_t}\boldsymbol{y}_t^\top$;
4   **end**
5   **for** $t \leftarrow 1$ to $T-1$ *in parallel* **do**
6     $\frac{\partial\ell}{\partial\boldsymbol{N}_t} \leftarrow -\frac{\partial\ell}{\partial\boldsymbol{\beta}_{t+1}}\boldsymbol{y}_t^\top - \boldsymbol{y}_{t+1}\frac{\partial\ell}{\partial\boldsymbol{\beta}_t^\top}$;
7   **end**

---

**Algorithm 4:** Substitute

**Input:** $\boldsymbol{L}_{1:T}, \boldsymbol{P}_{1:T-1}, \boldsymbol{\alpha}_{1:T}$
**Output:** $\boldsymbol{\alpha}_{1:T}$
1   **for** $t \leftarrow 2$ to $T$ **do**
     /* forward substitute */
2     $\boldsymbol{\alpha}_t \leftarrow \boldsymbol{\alpha}_t - \boldsymbol{P}_{t-1}\boldsymbol{\alpha}_{t-1}$;
3   **end**
4   **for** $t \leftarrow 1$ to $T$ *in parallel* **do**
5     $\boldsymbol{\alpha}_t \leftarrow \boldsymbol{L}_t^{-\top}\left(\boldsymbol{L}_t^{-1}\boldsymbol{\alpha}_t\right)$;
6   **end**
7   **for** $t \leftarrow T-1$ to $1$ **do**
     /* backward substitute */
8     $\boldsymbol{\alpha}_t \leftarrow \boldsymbol{\alpha}_t - \boldsymbol{P}_t^\top\boldsymbol{\alpha}_{t+1}$;
9   **end**

---

**Numerical Stability Considerations.** An important aspect of our solver design is the numerical stability offered by the direct method of Cholesky decomposition compared to iterative methods like the conjugate gradient (CG) algorithm. Both direct and iterative methods have errors influenced by the condition number $\kappa$ of the matrix. However, direct methods tend to be more stable in practice

because they compute an exact solution up to machine precision. In contrast, CG is iterative and can suffer from error accumulation across iterations, especially when the matrix is ill-conditioned or when the number of iterations is limited for computational reasons. This is crucial in our applications, where accurate solutions are necessary for modeling chaotic systems.

**Complexity Analysis.** The original MNN *dense* (exact) solver described in Pervez et al. (2024) has time complexity $O\left(T^3V^3R^3\right)$ and space complexity $O\left(T^2V^2R^2\right)$, for time $T$, $V$ variables, and $R$ derivative orders. The original *sparse* (approximate) solver working via conjugate gradient has both time and space complexities $O\left(T^2V^2R^2\right)$. In our proposed exact S-MNN, the time complexity is reduced by a $\Theta(T^2)$ factor to $O\left(TV^3R^3\right)$ from the MNN *dense* solver, and the space complexity is reduced to $O\left(TV^2R^2\right)$. Thus, both time and memory requirements now depend *linearly* on the number of time points $T$, making the new method more scalable for longer trajectories.

## 4 Related Work in Scientific Machine Learning

Scientific machine learning has emerged as a transformative field that combines data-driven approaches with domain-specific knowledge to model complex dynamical systems. Various specialized methodologies have been developed to tackle different aspects of this challenge, particularly in solving differential equations using neural networks and, to a lesser extent, inverse problems.

**Models for Prediction.** Neural Ordinary Differential Equations (Neural ODEs) (Chen et al., 2018; 2021; Kidger et al., 2021; Norcliffe et al., 2020) model continuous-time dynamics by parameterizing the derivative of the hidden state with a neural network. Neural ODEs, however, are constrained by the structure and sequential nature of ODE solvers and can be inefficient to train. Neural Operators (Li et al., 2020b;a; 2024; 2021; Azizzadenesheli et al., 2024; Boullé & Townsend, 2023) are designed to learn mappings between infinite-dimensional function spaces, enabling the modeling of PDEs and complex spatial-temporal patterns. However, their focus on lower frequencies in the Fourier spectrum can lead to poor prediction over longer roll-outs (Lippe et al., 2023).

**Models for Discovery.** The line of work on Sparse Identification of Nonlinear Dynamical Systems (SINDy) (Kaheman et al., 2020; Brunton et al., 2016b;a; Kaptanoglu et al., 2021; Course & Nair, 2023; Lu et al., 2022; Rudy et al., 2017) aims to discover governing equations by identifying sparse representations within a *predefined* library of candidate functions. However, SINDy is only a generalized linear model that does not use neural networks and can struggle with highly complex, noisy, or strongly nonlinear systems. Physics-informed networks and universal differential equations (Raissi et al., 2019; Rackauckas et al., 2020) also work as discovery methods for inferring unknown terms in PDEs. Symbolic regression methods (Udrescu & Tegmark, 2020; d'Ascoli et al., 2023) constitute another line of work that aims to discover purely symbolic expressions from data.

**Discussion.** Mechanistic Neural Networks (MNNs) (Pervez et al., 2024) have been proposed as a single framework for prediction and discovery. MNNs compute ODE representations from data which provide a strong inductive bias for scientific ML tasks. However, MNN training introduces significant challenges that require solving large linear systems during both the forward and backward passes and demands substantial computational resources. Our method addresses this scalability problem by reducing the computational complexity and enables applications on long sequences.

## 5 Experiments

To demonstrate the effectiveness and scalability of our proposed *Scalable* Mechanistic Neural Network (S-MNN), we conduct experiments across multiple settings in scientific machine learning applications for dynamical systems including governing equation discovery for the Lorenz system (Section 5.2), solving the Korteweg-de Vries (KdV) partial derivative equation (PDE) (Section 5.3), and sea surface temperature (SST) prediction for modeling long real-world temporal sequences (Section 5.4). We show that S-MNN matches the precisions and convergence rates of the original MNN (Pervez et al., 2024) while significantly reducing computational time and GPU memory usage. We also compare S-MNN with other state-of-the-art methods in these experiments.

## 5.1 STANDALONE VALIDATION

To assess the correctness of our solver in solving linear ordinary differential equations (ODEs), we conduct a standalone validation. Our solver is designed to solve linear ODEs directly (Section 3.1) without incorporating additional neural network layers or trainable parameters.

**Experiment Settings.** We select five linear ODE problems from ODEBench (d'Ascoli et al., 2024)—RC Circuit, Population Growth, Language Death Model, Harmonic Oscillator, and Harmonic Oscillator with Damping—that are commonly used in various scientific fields such as physics and biology, along with an additional third-order ODE. Mathematical details about these ODEs are provided in Appendix B.1. For each problem, we discretize the time axis into 1,000 steps with a uniform step size of 0.01 and apply our S-MNN solver.

**Results and Discussion.** We compare the numerical solutions obtained by our solver against the corresponding closed-form solutions. Figure 2 presents the results, where we plot the solutions $y(t)$ along with their first and second derivatives $y'(t)$ and $y''(t)$. The numerical results from our solver closely match the analytical solutions, exhibiting negligible differences. These results confirm that our solver is capable of correctly solving linear ODEs. In Appendix B.1, Table 3, we provide the exact errors for each benchmark, and the comparisons to the classic solvers RK45 (Dormand & Prince, 1980; Shampine, 1986) and LSODA (Hindmarsh, 1983; Petzold, 1983).

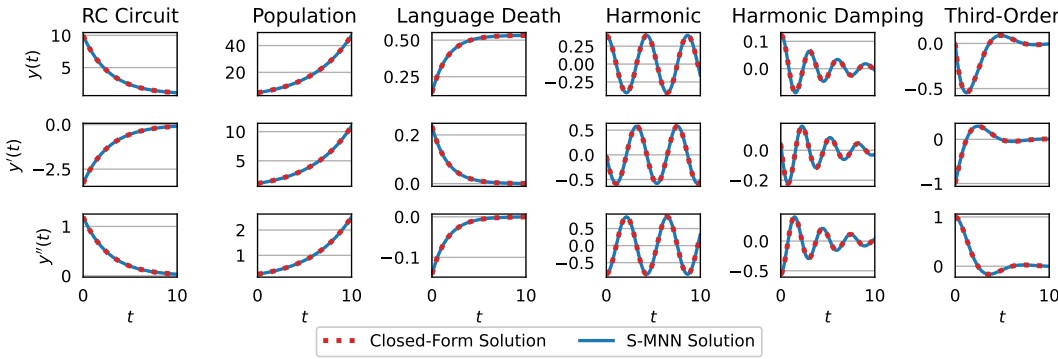

Figure 2: Standalone S-MNN solver validation results compared with the closed-form solutions.

## 5.2 COMPARATIVE ANALYSIS: DISCOVERY OF GOVERNING EQUATIONS

In this experiment, we evaluate the capability of our S-MNN in discovering the coefficients of the governing equations for the Lorenz system following Section 5.1 in the origin MNN paper (Pervez et al., 2024). The Lorenz system is a set of nonlinear ODEs known for its chaotic behavior, making it a standard benchmark for testing equation discovery methods in dynamical systems. The governing equations are given by

$$\begin{cases} \mathrm{d}x/\mathrm{d}t = \sigma\,(y - x) = a_1 x + a_2 y \\ \mathrm{d}y/\mathrm{d}t = x\,(\rho - z) - y = a_3 x + a_4 y + a_5 xz \\ \mathrm{d}z/\mathrm{d}t = xy - \beta z = a_6 z + a_7 xy \end{cases} \quad (15)$$

where $a_1, \ldots, a_7 \in \mathbb{R}$ are the coefficients to be discovered.

**Dataset.** The dataset is generated by numerically integrating the Lorenz system equations using the standard parameters $\sigma = 10$, $\rho = 28$, and $\beta = 8/3$. We use the initial condition $x = y = z = 1$ and integrate over 10,000 time steps with a step size of 0.01 using the `scipy.integrate.odeint` function from SciPy.

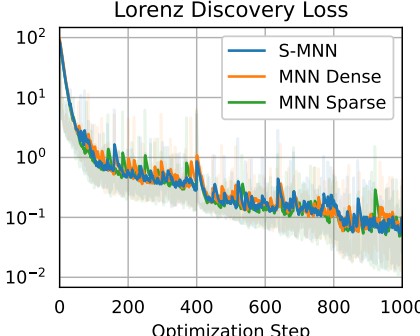

Figure 3: Lorenz discovery loss over first 1,000 optimization steps (exponential moving average factor = 0.9) using S-MNN (ours) compared with the original MNN *dense* and *sparse* solvers (Pervez et al., 2024).

**Experiment Settings.** We apply our solver to the same network architecture and dataset in Pervez et al. (2024). We train the model with the default settings: sequence length of 50 and batch size of 512. An instance in the batch is created by sampling a random chunk from the trajectory. Please check Appendix B.2 for more details about the batches. The training objective is to minimize the difference between the predicted and true trajectories by optimizing the coefficients $a_1, \ldots, a_7$. Then, to assess the scalability of our method, we measure the runtime and the GPU memory consumption across different sequence lengths and batch sizes using an NVIDIA H100 GPU with 80 GB of memory.

**Results and Discussion.** Figure 3 illustrates the loss convergence over the optimization steps for our S-MNN solver compared to the original MNN *dense* and *sparse* solvers. The S-MNN achieves similar convergence rates, confirming that the removal of slack variables does not compromise accuracy. The final discovered coefficients $a_0, \ldots, a_7$ are presented in Appendix B.2, Table 4, alongside results from the state-of-the-art method SINDy (Kaheman et al., 2020) for reference.

Table 1 summarizes the performance and GPU memory usage. Our S-MNN solver not only maintains high accuracy but also offers substantial efficiency improvements. Specifically, compared to the MNN *dense* solver, our method achieves a 4.9× speedup and reduces GPU memory usage by 50% for the default setting (batch size 512, sequence length 50). The more significant improvement in runtime compared to memory usage is expected, as our approach reduces runtime from $O\left(T^3\right)$ to $O\left(T\right)$, and memory from $O\left(T^2\right)$ to $O\left(T\right)$, with $T$ denoting the sequence length. Our S-MNN solver maintains high performance even with larger batch sizes and sequence lengths where the MNN solvers run out of memory or become computationally infeasible.

Table 1: Performance and GPU memory usage comparisons between MNN (Pervez et al., 2024) and S-MNN (ours) for the Lorenz discovery experiment.

| Batch Size | | **512** | **512** | **512** | 64 | 4096 |
|---|---|---|---|---|---|---|
| Sequence Length | | **50** | 5 | 500 | **50** | **50** |
| | MNN Dense | 36.4 | 9.5 | N/A[1] | 10.0 | 208.1 |
| Time per Optimization Step [ms] | MNN Sparse | 104.4 | 76.5 | >589.7[2] | 80.5 | 406.8 |
| | *S-MNN* | **7.4** | **5.5** | **32.2** | **5.5** | **18.3** |
| | MNN Dense | 2.77 | 1.18 | >80.00[1] | 1.33 | 14.85 |
| GPU Memory Usage [GiB] | MNN Sparse | 1.69 | **0.93** | 9.83[2] | **0.96** | 7.96 |
| | *S-MNN* | **1.38** | 1.32 | **1.96** | 1.33 | **1.81** |

[1] Out of memory error. [2] Loss does not converge after a large number (200) of conjugate gradient iterations.

### 5.3 COMPARATIVE ANALYSIS: SOLVING PARTIAL DIFFERENTIAL EQUATIONS (PDEs)

Next, we evaluate the capability of our S-MNN in solving partial differential equations (PDE), specifically focusing on the Korteweg-De Vries (KdV) equation, which is a third-order nonlinear PDE that describes the propagation of waves in shallow water and is expressed as

$$\frac{\partial y}{\partial t} + \frac{\partial^3 y}{\partial x^3} - 6y\frac{\partial y}{\partial x} = 0, \tag{16}$$

where $y\left(x, t\right)$ represents the wave amplitude as a function of spatial coordinate $x$ and time $t$. Solving the KdV equation is challenging due to its nonlinearity and the involvement of higher-order spatial derivatives, making it a popular benchmark for PDEs.

**Dataset.** We consider the KdV dataset provided by Brandstetter et al. (2022). The dataset consists of 512 samples each for training, validation, and testing. Each sample has a spatial domain of 256 meters and a temporal domain of 140 seconds, discretized into 256 spatial points and 140 temporal steps.

**Experiment Settings.** We model the temporal evolution at each spatial point as an independent ODE. A ResNet-1D architecture (Brandstetter et al., 2022) is employed to encode the temporal and spatial dependencies in the input sequences and feed them into the mechanistic solver. The sequence length is set to 10 seconds, and the model is trained to predict the wave profile over the next 9 seconds. The model is trained for 800 epochs. We repeat the same experiment for the original MNN *dense* and *sparse* solvers as well as our S-MNN solver.

**Results and Discussion.** Following Brandstetter et al. (2022), we report the testing error using the rollout averaged normalized mean squared error (NMSE), defined as

$$NMSE = \frac{1}{T} \sum_{t=1}^{T} \frac{\sum_x \left(y\left(x,t\right) - \hat{y}\left(x,t\right)\right)^2}{\sum_x \hat{y}^2\left(x,t\right)} \tag{17}$$

where $y\left(x,t\right)$ is the ground truth and $\hat{y}\left(x,t\right)$ is the model output. Table 2 presents the NMSE results for MNN and S-MNN, along with ResNet and FNO results taken from Brandstetter et al. (2022). The results indicate that both S-MNN and the MNN *dense* solver significantly outperform the ResNet and FNO models. Our S-MNN solver achieves slightly better accuracy than the MNN *dense* solver. The MNN *sparse* solver fails to converge in this experiment. We also provide visualizations for 100-second predictions in Appendix B.3, Figure 6.

Table 2: KdV prediction error (NMSE) for ResNet (Brandstetter et al., 2022), FNO (Brandstetter et al., 2022), MNN (Pervez et al., 2024), and S-MNN (ours). The errors are calculated on a 20-second prediction sequence unless otherwise stated.

| Method | NMSE | Method | NMSE | Method | NMSE |
|---|---|---|---|---|---|
| ResNet | 0.0223 | FNO | 0.0276 | MNN Sparse | No Convergence |
| ResNet-LPSDA-1 | 0.0200 | FNO-LPSDA | 0.0055 | MNN Dense | 0.00006 |
| ResNet-LPSDA-2 | 0.0111 | FNO-AR | 0.0030 | MNN Dense | 0.00032 (40 sec) |
| ResNet-LPSDA-3 | 0.0155 | FNO-AR-LPSDA | 0.0010 | *S-MNN* | **0.00005** |
| ResNet-LPSDA-4 | 0.0113 | | | *S-MNN* | **0.00037** (40 sec) |

In terms of computational performance, the training time for S-MNN is significantly reduced to 10.1 hours compared to 38.0 hours for the MNN *dense* solver, indicating a substantial speedup. Additionally, our method consumes less GPU memory, using 2.19 GiB versus 3.40 GiB for the original solver. The MNN *sparse* solver does not converge within a reasonable time frame, taking 82.4 hours and 3.07 GiB without achieving satisfactory results.

## 5.4 REAL-WORLD APPLICATION: LONG-TERM SEA SURFACE TEMPERATURE FORECASTING

The ability to handle longer sequences and larger batch sizes without sacrificing performance positions our S-MNN as a powerful tool for modeling complex dynamical systems. In this section, we demonstrate a real-world example use case: sea surface temperature (SST) prediction. SST exhibits long periodic features that can only be effectively captured with long sequences.

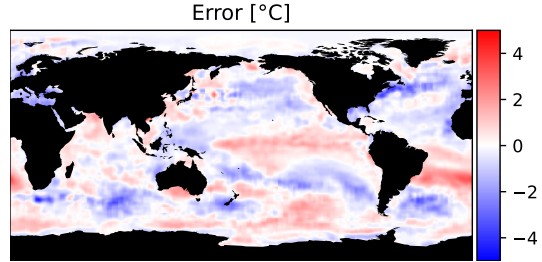

Figure 4: Error visualization for the S-MNN 4-year sea surface temperature (SST) prediction.

**Dataset.** We use the SST-V2 dataset (Huang et al., 2021), which provides weekly mean sea surface temperatures for 1,727 weeks from December 31, 1989, to January 29, 2023, over a 1° latitude × 1° longitude global grid (180 × 360).

**Experiment Settings.** We employ S-MNN and MNN with the Mechanistic Identifier proposed by Yao et al. (2024) to predict SST data. The model leverages mechanistic layers to capture the underlying dynamics of SST. We set the default batch size to 12,960 (corresponding to 6,480 pairs of grid points and their randomly selected neighboring points) and the sequence length (chunk length) to 208 weeks. The dataset is split so that the latest chunk of measurements is reserved for testing while the remaining data is used for training. The model is trained for 1,000 epochs. To evaluate the scalability and stability, we benchmark the model with different sequence lengths. Besides S-MNN and MNN, we also conduct the experiment using Ada-GVAE (Locatello et al., 2020), which is also used as a baseline in Yao et al. (2024).

**Results and Discussion.** Figure 1 visualizes the 4-year (208-week) prediction made by our S-MNN with the Mechanistic Identifier. Figure 4 visualizes its prediction error over the ground truth. The S-MNN effectively captures the spatial patterns of SST, demonstrating high predictive precision.

To quantitatively assess the performance and scalability, we measure the accuracy in terms of the relative MSE (mean squared error over the standardized data), as well as the runtime and GPU memory usage across different sequence lengths. Figure 5 summarizes these results.

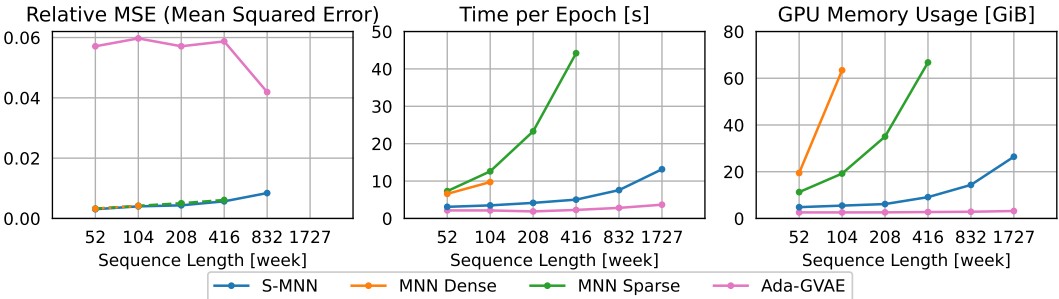

Figure 5: Testing error, training runtime and GPU memory usage comparisons between S-MNN (ours), MNN *dense* and *sparse* solvers (Pervez et al., 2024), and Ada-GVAE (Locatello et al., 2020) for SST forecasting. Note that the x-axis is in log scale, so both runtime and memory consumption of S-MNN increase linearly as expected.

We observe that the relative MSE averaged over both sequence length and batch size remains low for both the MNN and S-MNN solvers, with our S-MNN maintaining low MSE even for longer sequences where the MNN solvers cannot run due to resource limitations. A slight increase in the relative MSE for longer sequences is expected, as modeling longer-term dependencies introduces increased complexity, and the error accumulates over extended prediction horizons. Note that for the sequence length of 1,727, we use the entire dataset for training, leaving no separate testing dataset for evaluation, which is why no MSE results can be provided in Figure 5. Given the log scale in the x-axis, we observe that S-MNN demonstrates a linear increase in both runtime and memory consumption with respect to sequence length, aligning with our theoretical complexities for time and space. In contrast, the MNN solvers exhibit much steeper increases in runtime and memory, and the memory usage quickly exceeds the 80 GiB limit of our GPU for longer sequences. We are unable to run the MNN *dense* solver for sequence lengths beyond 104 weeks and the *sparse* solver beyond 416 weeks.

## 6 CONCLUSION

This paper introduces the *Scalable* Mechanistic Neural Network (S-MNN), addressing the scalability limitations of the original Mechanistic Neural Networks (Pervez et al., 2024) (MNN) by reducing computational complexities to *linear* in sequence length for both time and space. This is achieved by eliminating slack variables and central difference constraints, transitioning from quadratic programming to least squares regression, and exploiting banded matrix structures within the solver. Our experiments demonstrate that S-MNN retains the precision of the original MNN while significantly enhancing computational efficiency. Given these substantial advantages, S-MNN can drop-in replace the original MNN. We believe this advancement can provide a practical and efficient method for embedding mechanistic knowledge into neural network models for complex dynamical systems.

**Limitations and Future Work.** While our S-MNN significantly enhances scalability, certain components, such as the sequential for-loops in the Cholesky decomposition (Algorithm 3) and the forward/backward substitution steps (Algorithm 4), still limit parallelism along the time dimension due to inherent data dependencies. This sequential execution can become a bottleneck when the batch and block sizes are small compared to the number of time steps, leading to underutilization of GPU resources and increased CPU overhead from launching small GPU kernels. Although we have employed CUDA Graphs to reduce this overhead, the fundamental sequential nature of the algorithms remains unaddressed. For future work, we aim to develop algorithms that retain linear time and space complexities but introduce full parallelism also across the time dimension.

ETHICS STATEMENT

Throughout this work, we have strictly adhered to the ICLR Code of Ethics. All datasets utilized in our experiments are publicly available and widely recognized within the scientific community. We ensure that these datasets do not contain any personally identifiable information or sensitive content. Our work does not involve human subjects, animals, or any form of personal data collection. We have thoroughly considered potential dual-use concerns and do not foresee any harmful applications of our methods. There are no conflicts of interest to declare, and no external sponsorship influenced the outcomes of this research. All experiments were conducted with integrity and transparency.

REPRODUCIBILITY STATEMENT

We are committed to ensuring that our work is transparent and reproducible. To facilitate this, we share the source code of both our S-MNN solver and experiments as part of the supplementary materials. The code is documented and includes instructions for setting up the environment, running the simulations, and reproducing the results presented in our paper. By making our resources openly available and providing detailed explanations, we aim to enable the research community to validate and build upon our findings.

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

# A THEORETICAL DERIVATIONS

## A.1 DEFINITIONS OF $A$, $b$, $W$, AND $y$

$A$, $b$, $W$, and $y$ are defined as follows. $A$, $b$, and $W$ are only theoretical and they are not explicitly constructed during computation. Only $y$ is computed.

Let $\bar{\bar{\bar{A}}}_{t,q,v} = [c_{t,q,v,0}, \ldots, c_{t,q,v,R}]^\top \in \mathbb{R}^{R+1}$.

Let $\bar{\bar{A}}_{t,q} = \left[\bar{\bar{\bar{A}}}_{t,q,1}^\top, \ldots, \bar{\bar{\bar{A}}}_{t,q,V}^\top\right]^\top \in \mathbb{R}^{V(R+1)}$.

Let $\bar{A}_t = \left[\bar{\bar{A}}_{t,1}, \ldots, \bar{\bar{A}}_{t,Q}\right]^\top \in \mathbb{R}^{Q \times V(R+1)}$.

Let $A_{\text{gov}} = \text{Diag}\left(\bar{A}_1, \ldots, \bar{A}_T\right) \in \mathbb{R}^{TQ \times n}$.

Let $\tilde{\tilde{A}}_{\text{init}} = [I, 0] \in \mathbb{R}^{(R_{\text{init}}+1) \times (R+1)}$.

Let $\tilde{A}_{\text{init}} = \text{Diag}(\underbrace{\tilde{\tilde{A}}_{\text{init}}, \ldots, \tilde{\tilde{A}}_{\text{init}}}_{T_{\text{init}}V \text{ times}}) \in \mathbb{R}^{T_{\text{init}}V(R_{\text{init}}+1) \times T_{\text{init}}V(R+1)}$.

Let $A_{\text{init}} = \left[\tilde{A}_{\text{init}}, 0\right] \in \mathbb{R}^{T_{\text{init}}V(R_{\text{init}}+1) \times n}$.

Let $\hat{A}_t^+ \in \mathbb{R}^{(R+1) \times (R+1)}$ such that $\left[\hat{A}_t^+\right]_{i,j} = \begin{cases} 0 & \text{if } i > j, \\ s_t^{j-i}/(j-i)! & \text{otherwise.} \end{cases}$

Let $\hat{A}_t^- \in \mathbb{R}^{(R+1) \times (R+1)}$ such that $\left[\hat{A}_t^-\right]_{i,j} = \begin{cases} 0 & \text{if } i > j, \\ (-s_t)^{j-i}/(j-i)! & \text{otherwise.} \end{cases}$

Let $A_t^+ = \text{Diag}(\underbrace{\hat{A}_t^+, \ldots, \hat{A}_t^+}_{V \text{ times}}) \in \mathbb{R}^{V(R+1) \times V(R+1)}$.

Let $A_t^- = \text{Diag}(\underbrace{\hat{A}_t^-, \ldots, \hat{A}_t^-}_{V \text{ times}}) \in \mathbb{R}^{V(R+1) \times V(R+1)}$.

Let $A_{\text{smooth\_forward}} = \begin{bmatrix} A_1^+ & -I & & \\ & \ddots & \ddots & \\ & & A_{T-1}^+ & -I \end{bmatrix} \in \mathbb{R}^{(T-1)V(R+1) \times n}$.

Let $A_{\text{smooth\_backward}} = \begin{bmatrix} -I & A_1^- & & \\ & \ddots & \ddots & \\ & & -I & A_{T-1}^- \end{bmatrix} \in \mathbb{R}^{(T-1)V(R+1) \times n}$.

Let

$$A = \begin{bmatrix} A_{\text{gov}} \\ A_{\text{init}} \\ A_{\text{smooth\_forward}} \\ A_{\text{smooth\_backward}} \end{bmatrix} \in \mathbb{R}^{m \times n}. \tag{18}$$

Let $\tilde{b}_t = [d_{t,1}, \ldots, d_{t,Q}]^\top \in \mathbb{R}^Q$.

Let $b_{\text{gov}} = \left[\tilde{b}_1^\top, \ldots, \tilde{b}_T^\top\right]^\top \in \mathbb{R}^{TQ}$.

Let $\bar{\bar{b}}_{t,v} = [u_{t,v,0}, \ldots, u_{t,v,R_{\text{init}}}]^\top \in \mathbb{R}^{R_{\text{init}}+1}$.

Let $\bar{b}_t = \left[\bar{\bar{b}}_{t,1}^\top, \ldots, \bar{\bar{b}}_{t,V}^\top\right]^\top \in \mathbb{R}^{V(R_{\text{init}}+1)}$.

Let $\boldsymbol{b}_{\text{init}} = \left[\bar{\boldsymbol{b}}_1^\top, \ldots, \bar{\boldsymbol{b}}_{T_{\text{init}}}^\top\right]^\top \in \mathbb{R}^{T_{\text{init}} V (R_{\text{init}}+1)}$.

Let
$$\boldsymbol{b} = \left[\boldsymbol{b}_{\text{gov}}^\top, \boldsymbol{b}_{\text{init}}^\top, \boldsymbol{0}^\top\right]^\top \in \mathbb{R}^m. \tag{19}$$

Let $\boldsymbol{w}_{\text{gov}} = [\underbrace{w_{\text{gov}}^2, \ldots, w_{\text{gov}}^2}_{TQ \text{ times}}]^\top \in \mathbb{R}^{TQ}$.

Let $\boldsymbol{w}_{\text{init}} = [\ \underbrace{w_{\text{init}}^2, \ldots, w_{\text{init}}^2}_{T_{\text{init}} V (R_{\text{init}}+1) \text{ times}}\ ]^\top \in \mathbb{R}^{T_{\text{init}} V (R_{\text{init}}+1)}$.

Let $\bar{\bar{\boldsymbol{w}}}_t = \left[\left(w_{\text{smooth}} s_t^0\right)^2, \ldots, \left(w_{\text{smooth}} s_t^R\right)^2\right]^\top \in \mathbb{R}^{R+1}$.

Let $\bar{\boldsymbol{w}}_t = [\underbrace{\bar{\bar{\boldsymbol{w}}}_t^\top, \ldots, \bar{\bar{\boldsymbol{w}}}_t^\top}_{V \text{ times}}]^\top \in \mathbb{R}^{V (R_{\text{init}}+1)}$.

Let $\boldsymbol{w}_{\text{smooth}} = \left[\bar{\boldsymbol{w}}_1^\top, \ldots, \bar{\boldsymbol{w}}_{T-1}^\top\right]^\top \in \mathbb{R}^{(T-1) V (R+1)}$.

Let $\boldsymbol{w} = \left[\boldsymbol{w}_{\text{gov}}^\top, \boldsymbol{w}_{\text{init}}^\top, \boldsymbol{w}_{\text{smooth}}^\top, \boldsymbol{w}_{\text{smooth}}^\top\right]^\top \in \mathbb{R}^m$.

Let
$$\boldsymbol{W} = \text{diag}\left(\boldsymbol{w}\right) \in \mathbb{R}^{m \times m}. \tag{20}$$

Let $\bar{\bar{\boldsymbol{y}}}_{t,v} = [y_{t,v,0}, \ldots, y_{t,v,R}]^\top \in \mathbb{R}^{R+1}$.

Let $\bar{\boldsymbol{y}}_t = \left[\bar{\bar{\boldsymbol{y}}}_{t,1}^\top, \ldots, \bar{\bar{\boldsymbol{y}}}_{t,V}^\top\right]^\top \in \mathbb{R}^{V (R+1)}$.

Let
$$\boldsymbol{y} = \left[\bar{\boldsymbol{y}}_1^\top, \ldots, \bar{\boldsymbol{y}}_T^\top\right]^\top \in \mathbb{R}^n. \tag{21}$$

## A.2 COMPUTE $\boldsymbol{M}$ AND $\boldsymbol{\beta}$

Define matrix $\boldsymbol{C}_t \in \mathbb{R}^{Q \times V (R+1)}$ such that $[\boldsymbol{C}_t]_{q,i} = c_{t,q,v,r}$ where $v = \lfloor (i-1) / (R+1) \rfloor + 1$ and $r = (i-1) \mod (R+1)$.

Define vector $\boldsymbol{d}_t = [d_{t,1}, d_{t,2}, \ldots, d_{t,Q}]^\top \in \mathbb{R}^Q$.

Define constant matrix $\boldsymbol{U}_t \in \{0, 1\}^{V (R+1) \times V (R+1)}$ such that
$$[\boldsymbol{U}_t]_{i,j} = \begin{cases} 1 & \text{if } t \leq T_{\text{init}} \text{ and } ((i-1) \mod (R+1)) \leq R_{\text{init}} \text{ and } i = j, \\ 0 & \text{otherwise.} \end{cases} \tag{22}$$

Define vector $\boldsymbol{u}_t \in \mathbb{R}^{V (R+1)}$ such that
$$[\boldsymbol{u}_t]_i = \begin{cases} u_{t,v,r} & \text{if } t \leq T_{\text{init}} \text{ and } r \leq R_{\text{init}}, \\ 0 & \text{otherwise,} \end{cases} \tag{23}$$

where $v = \lfloor (i-1) / (R+1) \rfloor + 1$ and $r = (i-1) \mod (R+1)$.

Define constant matrix $\boldsymbol{F} \in \mathbb{R}^{(R+1) \times (R+1)}$ such that
$$[\boldsymbol{F}]_{i,j} = \begin{cases} 0 & \text{if } i > j, \\ 1 / (j-i)! & \text{otherwise.} \end{cases} \tag{24}$$

Define matrix $\boldsymbol{S}_t^+ = \text{diag}\left(s_t^0, s_t^1, \ldots, s_t^R\right) \in \mathbb{R}^{(R+1) \times (R+1)}$.

Define matrix $\boldsymbol{S}_t^- = \text{diag}\left((-s_t)^0, (-s_t)^1, \ldots, (-s_t)^R\right) \in \mathbb{R}^{(R+1) \times (R+1)}$.

Define matrix $\boldsymbol{S}_t^2 = \text{diag}\left(s_t^0, s_t^2, \ldots, s_t^{2R}\right) \in \mathbb{R}^{(R+1) \times (R+1)}$.

Define matrix $\boldsymbol{S}_t^* \in \mathbb{R}^{(R+1)\times(R+1)}$ such that

$$\boldsymbol{S}_t^* = \begin{cases} \left(\boldsymbol{S}_1^+\right)^\top \boldsymbol{F}^\top \boldsymbol{F} \boldsymbol{S}_1^+ + \boldsymbol{S}_1^2 & \text{if } t = 1, \\ \left(\boldsymbol{S}_{T-1}^-\right)^\top \boldsymbol{F}^\top \boldsymbol{F} \boldsymbol{S}_{T-1}^- + \boldsymbol{S}_{T-1}^2 & \text{if } t = T, \\ \left(\boldsymbol{S}_t^+\right)^\top \boldsymbol{F}^\top \boldsymbol{F} \boldsymbol{S}_t^+ + \left(\boldsymbol{S}_{t-1}^-\right)^\top \boldsymbol{F}^\top \boldsymbol{F} \boldsymbol{S}_{t-1}^- + \boldsymbol{S}_t^2 + \boldsymbol{S}_{t-1}^2 & \text{otherwise.} \end{cases} \tag{25}$$

Define matrix $\boldsymbol{S}_t^{**} = -\left(\boldsymbol{S}_t^+\right)^\top \boldsymbol{F} \boldsymbol{S}_t^+ - \left(\boldsymbol{S}_t^-\right)^\top \boldsymbol{F}^\top \boldsymbol{S}_t^- \in \mathbb{R}^{(R+1)\times(R+1)}$.

Define block diagonal matrix $\boldsymbol{S}_t = \text{Diag}(\underbrace{\boldsymbol{S}_t^*, \ldots, \boldsymbol{S}_t^*}_{V \text{ times}}) \in \mathbb{R}^{V(R+1)\times V(R+1)}$.

Then, $\boldsymbol{M}_t$, $\boldsymbol{N}_t$, and $\boldsymbol{\beta}_t$ can be computed as

$$\boldsymbol{M}_t = w_{\text{gov}}^2 \boldsymbol{C}_t^\top \boldsymbol{C}_t + w_{\text{init}}^2 \boldsymbol{U}_t + w_{\text{smooth}}^2 \boldsymbol{S}_t, \quad \forall t \in \{1, \ldots T\}, \tag{26}$$

$$\boldsymbol{N}_t = w_{\text{smooth}}^2 \text{Diag}(\underbrace{\boldsymbol{S}_t^{**}, \ldots, \boldsymbol{S}_t^{**}}_{V \text{ times}}), \quad \forall t \in \{1, \ldots T-1\}, \tag{27}$$

$$\boldsymbol{\beta}_t = w_{\text{gov}}^2 \boldsymbol{C}_t^\top \boldsymbol{d}_t + w_{\text{init}}^2 \boldsymbol{u}_t, \quad \forall t \in \{1, \ldots T\}. \tag{28}$$

### A.3 GRADIENTS OF $\boldsymbol{M}$ AND $\boldsymbol{\beta}$

In this proof, we use the subscripts $h, i, j, k$ to denote the element index, e.g., $y_k$ is the $k$-th component of vector $\boldsymbol{y}$, $M_{i,j}$ is the element in the $i$-th row and $j$-th column of matrix $\boldsymbol{M}$, and $M_{i,:}$ is the row vector in the $i$-th row of $\boldsymbol{M}$.

Rewrite $\boldsymbol{y} = \boldsymbol{M}^{-1}\boldsymbol{\beta}$ as

$$y_k = \sum_h \left[M^{-1}\right]_{k,h} \beta_h. \tag{29}$$

Differentiate $y_k$ with respect to $\beta_i$,

$$\frac{\partial y_k}{\partial \beta_i} = \left[M^{-1}\right]_{k,i}. \tag{30}$$

Differentiate $l$ with respect to $\beta_i$ using the chain rule,

$$\frac{\partial \ell}{\partial \beta_i} = \sum_k \frac{\partial \ell}{\partial y_k} \frac{\partial y_k}{\partial \beta_i} = \sum_k \left[M^{-1}\right]_{k,i} \frac{\partial \ell}{\partial y_k} = \left[\boldsymbol{M}^{-\top}\right]_{i,:} \frac{\partial \ell}{\partial \boldsymbol{y}}. \tag{31}$$

Express the derivative in vector form,

$$\frac{\partial \ell}{\partial \boldsymbol{\beta}} = \boldsymbol{M}^{-\top} \frac{\partial \ell}{\partial \boldsymbol{y}}. \tag{32}$$

Because $\boldsymbol{M}$ is symmetric, $\boldsymbol{M}^{-1}$ is also symmetric,

$$\frac{\partial \ell}{\partial \boldsymbol{\beta}} = \boldsymbol{M}^{-1} \frac{\partial \ell}{\partial \boldsymbol{y}}. \tag{33}$$

Differentiate $y_k$ with respect to $M_{i,j}$,

$$\frac{\partial y_k}{\partial M_{i,j}} = \frac{\partial \left(\sum_h \left[M^{-1}\right]_{k,h} \beta_h\right)}{\partial M_{i,j}} = \sum_h \beta_h \frac{\partial \left[M^{-1}\right]_{k,h}}{\partial M_{i,j}}. \tag{34}$$

Differentiate of $\left[M^{-1}\right]_{k,h}$ with respect to $M_{i,j}$,

$$\frac{\partial \left[M^{-1}\right]_{k,h}}{\partial M_{i,j}} = -\left[M^{-1}\right]_{k,i} \left[M^{-1}\right]_{j,h}. \tag{35}$$

Substitute the derivative back into the expression,

$$\frac{\partial y_k}{\partial M_{i,j}} = - \left[M^{-1}\right]_{k,i} \sum_h \left[M^{-1}\right]_{j,h} \beta_h = -\frac{\partial y_k}{\partial \beta_i} y_j. \tag{36}$$

Differentiate $l$ with respect to $M_{i,j}$ using the chain rule,

$$\frac{\partial \ell}{\partial M_{i,j}} = \sum_k \frac{\partial \ell}{\partial y_k} \frac{\partial y_k}{\partial M_{i,j}} = -y_j \sum_k \frac{\partial \ell}{\partial y_k} \frac{\partial y_k}{\partial \beta_i} = -y_j \frac{\partial \ell}{\partial \beta_i}. \tag{37}$$

Express the derivative in matrix form,

$$\frac{\partial \ell}{\partial \boldsymbol{M}} = -\frac{\partial \ell}{\partial \boldsymbol{\beta}} \boldsymbol{y}^\top. \tag{38}$$

### A.4 Overall Training and Testing Algorithms

Algorithms 5 and 6 present the training and testing procedures for a generic application employing our S-MNN framework. We utilize a trainable encoder $\theta_{\text{enc}}$, a trainable decoder $\theta_{\text{dec}}$, a training dataset $\mathcal{X}_{\text{train}}$, and a testing dataset $\mathcal{X}_{\text{test}}$. During training, the encoder and decoder are updated using gradient descent methods. In testing, we obtain the set of decoded outputs $\mathcal{Z}$ and the corresponding losses $\mathcal{L}$. The lines highlighted in yellow boxes are particularly pertinent to S-MNN. Specifically, during training, $\boldsymbol{L}_{1:T}$, $\boldsymbol{P}_{1:T-1}$, and $\boldsymbol{y}_{1:T}$ are computed in the forward pass and reused in the backward pass. In testing, these quantities are computed but discarded after subsequent computations, as they are not needed beyond that point. The word "AutoDiff" in Algorithm 5 is the abbreviation of automatic differentiation.

---

**Algorithm 5:** Train a Epoch

**Input:** $\theta_{\text{enc}}, \theta_{\text{dec}}, \mathcal{X}_{\text{train}}$
**Output:** $\theta_{\text{enc}}, \theta_{\text{dec}}$

1 **for** $\boldsymbol{x} \leftarrow$ a batch in $\mathcal{X}_{\text{train}}$ **do**
2      $c, d, u, s \leftarrow \theta_{\text{enc}}(\boldsymbol{x})$;
3      Compute $\boldsymbol{M}_{1:T}, \boldsymbol{N}_{1:T-1}, \boldsymbol{\beta}_{1:T}$ using $c, d, u, s$ ;
     // Appendix A.2
4      $\boldsymbol{L}_{1:T}, \boldsymbol{P}_{1:T-1}, \boldsymbol{y}_{1:T} \leftarrow$ SolverForwardPass $(\boldsymbol{M}_{1:T}, \boldsymbol{N}_{1:T-1}, \boldsymbol{\beta}_{1:T})$ ;
     // Algorithm 1, which calls Algorithms 3 and 4
5      $\boldsymbol{z} \leftarrow \theta_{\text{dec}}(\boldsymbol{y}_{1:T})$;
6      Compute loss $\ell$ using $\boldsymbol{z}$;
7      Compute gradient $\frac{\partial \ell}{\partial \boldsymbol{z}}$ via AutoDiff;
8      Compute gradients $\frac{\partial \ell}{\partial \theta_{\text{dec}}}, \frac{\partial \ell}{\partial \boldsymbol{y}_{1:T}}$ via AutoDiff;
9      $\frac{\partial \ell}{\partial \boldsymbol{M}_{1:T}}, \frac{\partial \ell}{\partial \boldsymbol{N}_{1:T-1}}, \frac{\partial \ell}{\partial \boldsymbol{\beta}_{1:T}} \leftarrow$ SolverBackwardPass $\left(\boldsymbol{L}_{1:T}, \boldsymbol{P}_{1:T-1}, \boldsymbol{y}_{1:T}, \frac{\partial \ell}{\partial \boldsymbol{y}_{1:T}}\right)$ ;
     // Algorithm 2, which calls Algorithm 4
10     Compute gradients $\frac{\partial \ell}{\partial c}, \frac{\partial \ell}{\partial d}, \frac{\partial \ell}{\partial u}, \frac{\partial \ell}{\partial s}$ via AutoDiff ;
11     Compute gradient $\frac{\partial \ell}{\partial \theta_{\text{enc}}}$ via AutoDiff;
12     Update $\theta_{\text{enc}}, \theta_{\text{dec}}$ using $\frac{\partial \ell}{\partial \theta_{\text{enc}}}, \frac{\partial \ell}{\partial \theta_{\text{dec}}}$;
13 **end**

---

---

**Algorithm 6:** Test

**Input:** $\theta_{\text{enc}}, \theta_{\text{dec}}, \mathcal{X}_{\text{test}}$
**Output:** $\mathcal{Z}, \mathcal{L}$

1 Initialize $\mathcal{Z}, \mathcal{L} \leftarrow \varnothing, \varnothing$;
2 **for** $x \leftarrow$ a batch in $\mathcal{X}_{\text{test}}$ **do**
3    $c, d, u, s \leftarrow \theta_{\text{enc}}(x)$;
4    Compute $M_{1:T}, N_{1:T-1}, \beta_{1:T}$ using $c, d, u, s$;
     // Appendix A.2
5    $L_{1:T}, P_{1:T-1}, y_{1:T} \leftarrow \text{SOLVERFORWARDPASS}(M_{1:T}, N_{1:T-1}, \beta_{1:T})$;
     // Algorithm 1, which calls Algorithms 3 and 4
6    $z \leftarrow \theta_{\text{dec}}(y_{1:T})$;
7    Compute loss $\ell$ using $z$;
8    $\mathcal{Z}, \mathcal{L} \leftarrow \mathcal{Z} \cup \{z\}, \mathcal{L} \cup \{\ell\}$
9 **end**

---

# B FURTHER EXPERIMENTAL DETAILS

## B.1 STANDALONE VALIDATION

We list the linear ODEs used for the validation experiment and their closed-form solutions. There are five ODEs from ODEBench (d'Ascoli et al., 2024) and an additional third-order ODE. $c_0, c_1, c_2$ are constant numbers. $u_0 = y(0), u_0 = y(0), u_1 = y'(0), u_2 = y''(0)$ are initial values.

RC-circuit (charging capacitor), $(c_0, c_1, c_2) = (0.7, 1.2, 2.31)$, $(u_0) = (10)$,

$$\frac{y}{c_1} + c_2 \frac{\mathrm{d}y}{\mathrm{d}t} = c_0, \tag{39}$$

$$y = c_0 c_1 + (u_0 - c_0 c_1) \exp\left(-\frac{t}{c_1 c_2}\right). \tag{40}$$

Population growth (naive), $(c_0) = (0.23)$, $(u_0) = (4.78)$,

$$c_0 y - \frac{\mathrm{d}y}{\mathrm{d}t} = 0, \tag{41}$$

$$y = u_0 \exp(c_0 t). \tag{42}$$

Language death model for two languages, $(c_0, c_1) = (0.32, 0.28)$, $(u_0) = (0.14)$,

$$(c_0 + c_1) y + \frac{\mathrm{d}y}{\mathrm{d}t} = c_0, \tag{43}$$

$$y = \frac{c_0}{c_0 + c_1} - \left(\frac{c_0}{c_0 + c_1} - u_0\right) \exp\left(-(c_0 + c_1) t\right). \tag{44}$$

Harmonic oscillator without damping, $(c_0) = (2.1)$, $(u_0, u_1) = (0.4, -0.03)$,

$$c_0 y + \frac{\mathrm{d}^2 y}{\mathrm{d}t^2} = 0, \tag{45}$$

$$y = u_0 \cos(t\sqrt{c_0}) + \frac{u_1}{\sqrt{c_0}} \sin(t\sqrt{c_0}). \tag{46}$$

Harmonic oscillator with damping, $(c_0, c_1) = (4.5, 0.43)$, $(u_0, u_1) = (0.12, 0.043)$,

$$c_0 y + c_1 \frac{\mathrm{d}y}{\mathrm{d}t} + \frac{\mathrm{d}^2 y}{\mathrm{d}t^2} = 0, \tag{47}$$

$$y = \exp\left(-\frac{c_1}{2} t\right) \left(u_0 \cos\left(\frac{t}{2}\sqrt{4c_0 - c_1^2}\right) + \frac{c_1 u_0 + 2u_1}{\sqrt{4c_0 - c_1^2}} \sin\left(\frac{t}{2}\sqrt{4c_0 - c_1^2}\right)\right). \tag{48}$$

Additional third-order ODE, $(u_0, u_1, u_2) = (0, -1, 1)$,

$$\frac{\mathrm{d}y}{\mathrm{d}t} + \frac{\mathrm{d}^2 y}{\mathrm{d}t^2} + \frac{\mathrm{d}^3 y}{\mathrm{d}t^3} = 0, \tag{49}$$

$$y = u_0 + u_1 + u_2 + \exp\left(-\frac{t}{2}\right)\left(-(u_1 + u_2)\cos\left(\frac{\sqrt{3}}{2}t\right) + \frac{\sqrt{3}}{3}(u_1 - u_2)\sin\left(\frac{\sqrt{3}}{2}t\right)\right). \tag{50}$$

In Table 3, we present the relative MSE (mean squared error over standardized data) and CPU runtime for RK45 (Dormand & Prince, 1980; Shampine, 1986), LSODA (Hindmarsh, 1983; Petzold, 1983), and our S-MNN method. RK45 and LSODA were evaluated using the `scipy.integrate.solve_ivp` function from SciPy. We run all solvers on CPUs (AMD EPYC 7513 32-Core Processor). This experiment serves as a sanity check to validate the correctness of our solver. S-MNN achieves a relative MSE below $10^{-6}$ in all cases, indicating excellent agreement with closed-form solutions. While S-MNN may not be the optimal solver for these pure ODE-solving tasks, it offers additional features, such as batched GPU processing and differentiability, which are not available in the classical solvers.

Table 3: Accuracy and performance comparisons between RK45 (Dormand & Prince, 1980; Shampine, 1986), LSODA (Hindmarsh, 1983; Petzold, 1983), and S-MNN (ours). The ODE problems are denoted using their initial letter: **R**C Circuit, **P**opulation, **L**anguage Death, **H**armonic, Harmonic **D**amping, and **T**hird-Order. Some high-order results are not applicable to low-order ODEs.

|  |  | Solver | R | P | L | H | D | T |
|---|---|---|---|---|---|---|---|---|
| Relative MSE | $y(t)$ | RK45 | 9.3e-12 | 2.9e-11 | 2.2e-11 | 1.7e-10 | 1.2e-10 | 6.7e-12 |
|  |  | LSODA | 4.5e-11 | 3.5e-10 | 9.2e-11 | 1.1e-09 | 1.5e-09 | 7.4e-11 |
|  |  | *S-MNN* | 4.8e-12 | 9.4e-12 | 2.6e-11 | 9.5e-08 | 2.1e-07 | 1.0e-09 |
|  | $y'(t)$ | RK45 | - | - | - | 1.5e-10 | 1.4e-10 | 4.4e-12 |
|  |  | LSODA | - | - | - | 7.6e-10 | 1.8e-09 | 5.9e-11 |
|  |  | *S-MNN* | 4.8e-12 | 9.4e-12 | 2.6e-11 | 7.7e-08 | 2.3e-07 | 6.8e-10 |
|  | $y''(t)$ | RK45 | - | - | - | - | - | 2.7e-12 |
|  |  | LSODA | - | - | - | - | - | 6.0e-11 |
|  |  | *S-MNN* | 2.2e-07 | 9.6e-08 | 7.1e-07 | 5.1e-07 | 4.0e-07 | 4.3e-09 |
| Runtime (CPU) [ms] |  | RK45 | 1.5 | 1.1 | 1.6 | 5.1 | 6.6 | 3.7 |
|  |  | LSODA | 1.1 | 0.9 | 1.3 | 5.0 | 5.0 | 2.9 |
|  |  | *S-MNN* | 22.2 | 22.2 | 22.1 | 23.8 | 23.8 | 26.1 |

## B.2 LORENZ DISCOVERY

Table 4: Discovered coefficients for the Lorenz system using SINDy (Kaheman et al., 2020), MNN (Pervez et al., 2024), and S-MNN (ours).

| Method | $a_1$ | $a_2$ | $a_3$ | $a_4$ | $a_5$ | $a_6$ | $a_7$ |
|---|---|---|---|---|---|---|---|
| Ground Truth | -10 | 10 | 28 | -1 | -1 | -8/3 | 1 |
| SINDy | -10.000 | 10.000 | 27.998 | -1.000 | -1.000 | -2.667 | 1.000 |
| MNN Dense | -10.0003 | 10.0003 | 27.9760 | -0.9934 | -0.9996 | -2.6660 | 0.9995 |
| MNN Sparse | -10.0055 | 10.0061 | 27.7165 | -0.9304 | -0.9937 | -2.6641 | 0.9990 |
| *S-MNN* | -10.0003 | 10.0004 | 27.9915 | -0.9968 | -0.9997 | -2.6664 | 1.0000 |

In the Lorenz system experiment, we generate a long trajectory starting from the initial condition $x = y = z = 1$ as the dataset. An instance in the batch is created by sampling a random chunk of this trajectory of length $T$. Each batch, therefore, consists of multiple such chunks, which can be thought of as sequences starting from different points along the trajectory. This approach effectively simulates having multiple sequences generated from different initial conditions, providing diversity in the training data. The default batch size of 512 ensures that the batch size is sufficiently large to

utilize the computational resources efficiently, mitigating the limitations associated with small batch and block sizes.

Table 4 lists the discovered coefficients after final fine-tuning, along with the results from the state-of-the-art method SINDy (Brunton et al., 2016a). Our S-MNN closely matches the ground truth, SINDy, and both the original MNN solvers, with only minor differences observed.

### B.3   KORTEWEG-DE VRIES (KDV) PREDICTION

Figure 6 illustrates a comparison between the ground truth solution, the prediction obtained from the original MNN *dense* solver, and that from our S-MNN. Both models produce predictions that closely align with the ground truth, demonstrating that our S-MNN effectively captures the intricate dynamics of the KdV equation. The MNN *sparse* solver cannot converge in this experiment and its result is not shown.

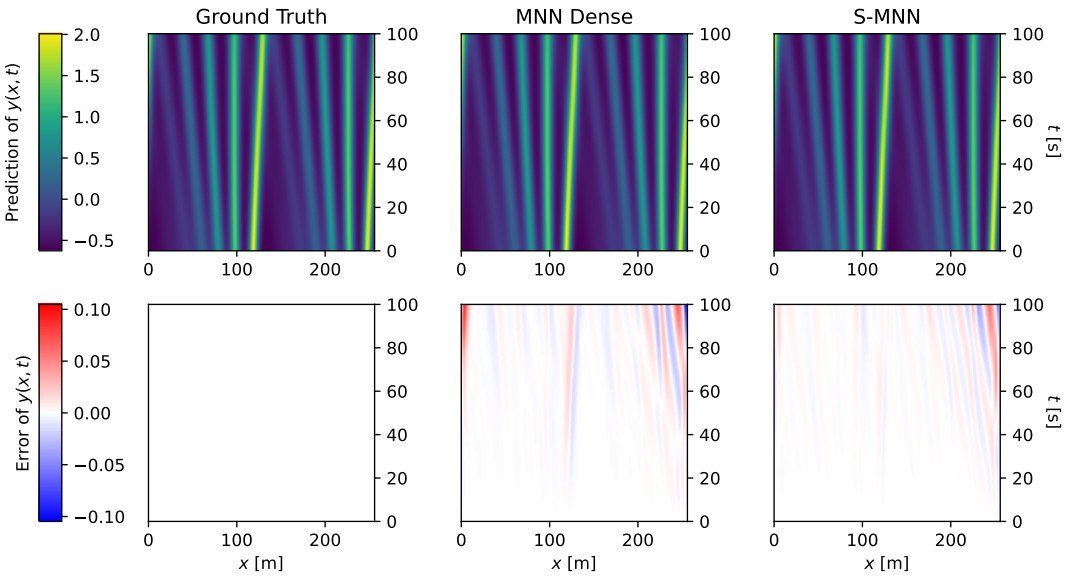

Figure 6: Visual comparisons between the ground truth, the MNN *dense* solver (Pervez et al., 2024), and our S-MNN solver for 100-second KdV predictions.

## C   COMPONENTS FROM THE ORIGINAL MECHANISTIC NEURAL NETWORK

The original Mechanistic Neural Network (MNN) (Pervez et al., 2024) solver approximates continuous-time dynamics through time discretization. Our modifications in S-MNN provide alternative approximation methods that improve efficiency without sacrificing accuracy. While our main focus is on presenting these improvements, for completeness, we briefly describe the components from the original MNN that we have modified or discarded. The symbols and notations used below are inherited from Section 3 and they may be different from Pervez et al. (2024).

### C.1   SLACK VARIABLES AND SMOOTHNESS CONSTRAINTS

Both S-MNN and MNN share the same constraints for the governing equations (Eq. 4) and initial values (Eq.5). However, unlike the smoothness constraints in S-MNN, which are formulated as equalities (Eqs. 6 and 7), the original MNN (Pervez et al., 2024) models the smoothness constraints as inequalities (Eqs. 54, 55, and 56): the approximation errors bounded by a slack variable $\epsilon \in \mathbb{R}$.

The forward and backward Taylor approximation errors in MNN are defined as:

$$E_{t,v,r}^{\text{forward}} = y_{t+1,v,r} - \sum_{r'=r}^{R} \frac{s_t^{r'-r}}{(r'-r)!} y_{t,v,r'}, \tag{51}$$

$$E_{t,v,r}^{\text{backward}} = y_{t,v,r} - \sum_{r'=r}^{R} \frac{(-s_t)^{r'-r}}{(r'-r)!} y_{t+1,v,r'}. \tag{52}$$

For the highest ($R$-th) order derivative, the central difference approximation error is used instead of the forward/backward approximation errors:

$$E_{t,v,r}^{\text{central}} = (y_{t+2,v,r} - y_{t,v,r}) - (s_t + s_{t+1}) y_{t+1,v,r+1}. \tag{53}$$

The smoothness constraints are expressed as inequalities involving the slack variable $\epsilon$:

$$s_t^r \left| E_{t,v,r}^{\text{forward}} \right| \leq \epsilon, \quad \forall t \in \{1, \ldots, T-1\}, \ v \in \{1, \ldots, V\}, \ r \in \{0, \ldots, R-1\}, \tag{54}$$

$$s_t^r \left| E_{t,v,r}^{\text{backward}} \right| \leq \epsilon, \quad \forall t \in \{1, \ldots, T-1\}, \ v \in \{1, \ldots, V\}, \ r \in \{0, \ldots, R-1\}, \tag{55}$$

$$(s_t + s_{t+1})^{R-2} \left| E_{t,v,R-1}^{\text{central}} \right| \leq \epsilon, \quad \forall t \in \{1, \ldots, T-2\}, \ v \in \{1, \ldots, V\}. \tag{56}$$

### C.2 Quadratic Programming Formulation

In the original MNN, the **approximation** problem is formulated as a linear programming (LP) problem: minimize $\epsilon$ while satisfying the linear equalities (Eqs. 4 and 5) and inequalities (Eqs. 54, 55, and 56). Specifically, the LP problem is:

$$\begin{aligned} \text{minimize} \quad & \epsilon \\ \text{subject to} \quad & \boldsymbol{A}' \boldsymbol{y}' \geq \boldsymbol{b}', \end{aligned} \tag{57}$$

where $\boldsymbol{A}' \in \mathbb{R}^{m' \times (n+1)}$, $\boldsymbol{b}' \in \mathbb{R}^{m'}$, $\boldsymbol{y}' = \left[\epsilon, \boldsymbol{y}^\top\right]^\top \in \mathbb{R}^{n+1}$, and $m'$ is the total number of constraints.

However, solving ODEs using LP within neural networks poses challenges: the solutions are not continuously differentiable, and efficient solvers are lacking. To address these issues, the LP problem is **relaxed** to a quadratic programming (QP) problem:

$$\begin{aligned} \text{minimize} \quad & \frac{\gamma}{2} \boldsymbol{y}'^\top \boldsymbol{y}' + \boldsymbol{\Delta}^\top \boldsymbol{y}' \\ \text{subject to} \quad & \boldsymbol{A}' \boldsymbol{y}' = \boldsymbol{b}', \end{aligned} \tag{58}$$

where $\boldsymbol{\Delta} = \left[\delta, \boldsymbol{0}^{1 \times n}\right]^\top \in \mathbb{R}^{n+1}$, and $\gamma, \delta \in \mathbb{R}$ are hyperparameters.

In QP (Eq. 58), $\epsilon^2$ is a minimized term. As an **approximation**, the absolute value operations $|\cdot|$ in Eqs. 54, 55, and 56 are dropped. The total number of constraints $m'$ becomes

$$m' = TQ + T_{\text{init}} V (R_{\text{init}} + 1) + 2 (T-1) VR + (T-2) V. \tag{59}$$

Using Lagrange multiplier $\boldsymbol{\lambda} \in \mathbb{R}^{m'}$, the solution can be found by solving the linear system:

$$\begin{bmatrix} \gamma \boldsymbol{I}^{(n+1) \times (n+1)} & \boldsymbol{A}'^\top \\ \boldsymbol{A}' & \boldsymbol{0}^{m' \times m'} \end{bmatrix} \begin{bmatrix} -\boldsymbol{y}' \\ \boldsymbol{\lambda} \end{bmatrix} = \begin{bmatrix} \boldsymbol{\Delta} \\ -\boldsymbol{b}' \end{bmatrix}. \tag{60}$$

However, the LP (Eq. 57) and QP (Eq. 58) problems are ill-defined because the number of constraints $m'$ exceeds the number of variables $n + 1$ when $T$ is large, making the problem infeasible. The square matrix in Eq. 60 is not full rank and the problem cannot be solved directly.

To circumvent this issue, the QP problem is transformed into its **dual** form:

$$\begin{bmatrix} \gamma \boldsymbol{I}^{m' \times m'} & \boldsymbol{A}' \\ \boldsymbol{A}'^\top & \boldsymbol{0}^{(n+1) \times (n+1)} \end{bmatrix} \begin{bmatrix} -\boldsymbol{\lambda} \\ \boldsymbol{y}' \end{bmatrix} = \begin{bmatrix} \boldsymbol{b}' \\ -\boldsymbol{\Delta} \end{bmatrix}. \tag{61}$$

The final solution for $\boldsymbol{y}'$ is then given by:

$$\boldsymbol{y}' = \left(\boldsymbol{A}'^{\top}\boldsymbol{A}'\right)^{-1}\left(\boldsymbol{A}'^{\top}\boldsymbol{b}' + \gamma\boldsymbol{\Delta}\right). \tag{62}$$

Notably, the QP solution of $\boldsymbol{y}'$ (Eq. 62) involves inverting the square matrix $\boldsymbol{M}' = \boldsymbol{A}'^{\top}\boldsymbol{A}'$ which is not a banded matrix due to the slack variable $\epsilon$. Consequently, solving this system has a time complexity of $O\left(T^3 V^3 R^3\right)$, where $T$ is the number of time steps, $V$ is the number of variables, and $R$ is the highest order of derivatives considered.

