# OpenReview forum: "Scalable Mechanistic Neural Networks"
_ICLR.cc/2025/Conference — ICLR 2025 Poster_

### Official Review · Reviewer_bGB5 · 2024-10-29

**Soundness:** 4
**Presentation:** 4
**Contribution:** 2
**Rating:** 6
**Confidence:** 4

**Summary:**

Building on mechanistic neural networks (MNN), this work proposes a strategy to efficiently solve long sequences using S-MNN. It addresses the scalability issues and reduces complexities to linear with 3 key improvements: elimination of slack variables, higher-order smoothness constraints, and least square regression. The result is a framework that can tackle long sequences with up to 4.9x speedup over MNN, and 50% GPU memory reduction, something the original MNN could not do.

**Strengths:**

Overall, the exposition is crisp, much clearer than the work it builds upon, of which the authors have given a great overview. The more direct formulation allows efficient forward pass, and a backward pass now directly supported by automatic differentiation. That said, the authors propose an analytical form that enables quicker numerical integration. The range of experiments is broad and the results are compelling, showing significant improvements over ordinary dense and sparse MNN.

**Weaknesses:**

1) The work does not address the trade-offs from the 3 main changes it brings, notably the consequence of dropping the slack variables. For instance, MNN notably allows **nonlinear** ODEs to be integrated with relatively ease. Is there any reason S-MNN doesn't do the same ?
2) Limited evaluation. The original MNN showed improved performance over ResNets and FNOs, but the current method has only been compared to MNN. Particularly, in this same setting, it would be interesting to know how well models outside the MNN family perform on the KdV and long-term sea surface temperature forecasting experiments, at the very least to gauge how difficult the task is.
3) Training and testing. During training, a number of quantities are learned. But during testing, it is not clear whether only the $y$s are solved for, and whether other matrix quantities from Algorithm 1-4 are reused. Appropriate training and testing algorithms should help clarify this.

**Questions:**

1) The paper mentions the batch size of 512 as a default setting. For the Lorentz system with initial conditions $x=y=z=1$, what makes up an instance in the "batch", especially since small batch and block sizes seems to be a limitation of S-MNN (line 536).
3) The Cholesky decomposition in Eq. 12 hinges on the matrix $M$ being positive definite, which is claimed in line 201. But how is that guaranteed ? The same remark goes for Eq. 28 where the symmetric nature of $M^{-1}$ is implied but not (re)stated.

---

> ### Author Response · Authors · 2024-11-21
> **Response to Reviewer bGB5 (1/2)**
>
> We thank you for the positive feedback. We are pleased that you found our exposition clear and the results compelling. We address your concerns and questions individually below.
>
> ***W1. Trade-offs from the Changes and Support for Nonlinear ODEs***
>
> * We would like to clarify that **S-MNN is designed as a drop-in replacement for the original MNN**, and it **supports nonlinear ODEs and other featured functionalities in exactly the same way** (e.g., use basis functions) **as MNN does**.
>
> * **In both MNN and S-MNN, nonlinear ODEs are handled by introducing auxiliary variables and incorporating them into the linear system.** Specifically, for each nonlinear term, we introduce auxiliary variables and add consistency terms to the loss function to ensure that these variables approximate the nonlinear functions of the solution. For more details, please refer to Section 3.4 in the original MNN paper \[1\].
>
> * **The original MNN solver is itself an approximation, and there are multiple ways to perform this approximation.** Regarding the trade-offs from the changes we made, it's important to note that due to the time-discretized nature of the problem, the solver inherently provides an approximation of the true continuous-time dynamics.  Our S-MNN approach uses alternative approximation methods, resulting in improved efficiency without sacrificing accuracy. **Essentially, the slack variables are artifacts of the original MNN approximation process, and by employing different approximation techniques, we can eliminate the need for them while retaining the model's capabilities.**
>
> ***W2. Evaluation and Comparison with Other Models***
>
> Our primary focus was on improving the computational efficiency of MNNs without compromising accuracy, demonstrating that S-MNN retains the performance of the original MNN while enabling applications to longer sequences and larger systems. The original MNN paper \[1\] provides comparisons with other baseline methods, such as ResNets and FNOs \[8\], showing that MNNs achieve competitive or superior performance in various tasks. Since S-MNN is designed to be a direct improvement over MNN, inheriting its strengths, we focused on comparing it against MNN to highlight the computational benefits.
>
> **We appreciate your suggestion and agree that including comparisons with models outside the MNN family would strengthen our work. We have included the following tables and figures in the revised manuscript of our submission.**
>
> * In Appendix B.1, Table 3, we provide the error and runtime for the solver validations (sanity checks) and the comparisons to the classic solvers RK45 \[9\] and LSODA \[10\]. S-MNN achieves excellent agreement with closed-form solutions. While S-MNN may not be the most efficient solver for these pure ODE-solving tasks, it offers additional features, such as batched GPU processing and differentiability, which are not available in the classical solvers.
>
> * In Appendix B.2, Table 4, we provide the discovered Lorenz coefficients and the comparisons to SINDy \[6\]. Our S-MNN closely matches SINDy, with only minor differences observed. Neural ODE \[3\] and FNO \[4\] do not do discovery tasks of this kind. This table is already in the initial submission referenced as Appendix B.2, Table 3\.
>
> * In Section 5.3, Table 2: We provide the normalized KdV prediction error (NMSE) and the comparisons to ResNet and FNO \[8\]. S-MNN significantly outperforms the ResNet and FNO models.
>
> * In Section 5.4, Figure 5: We provide the prediction error and performance for the sea surface temperature prediction task and the comparisons to the Ada-GVAE \[7\] (a plain MLP decoder), which was also used in \[2\] as a baseline. S-MNN has much smaller prediction errors than Ada-GVAE.
>
> **These results indicate S-MNN also significantly outperforms the listed baselines, inheriting the superior performance of MNN.**
>
> ***W3. Clarification on Training and Testing Procedures***
>
> During **testing (inference)**, only the variables ${y}$ are solved for. Since there is no backward pass during inference, the matrices and vectors involved in the solver (e.g., ${M}$, ${N}$, ${\\beta}$, ${L}$ and ${P}$) are computed as part of solving for ${y}$, but they are **not retained or reused** afterward. These quantities are generated on the fly during the forward pass and discarded once ${y}$ is obtained.
>
> We have updated the manuscript to include clearer descriptions and pseudo-codes of the training and testing procedures in **Appendix A.4** to enhance clarity.

---

> ### Author Response · Authors · 2024-11-21
> **Response to Reviewer bGB5 (2/2)**
>
> ***Q1. Batch in the Lorenz Experiment***
>
> In the Lorenz system experiment, **we generate a long trajectory starting from the initial condition** $x \= y \= z \= 1$. **An instance in the batch is created by sampling a random chunk** of this trajectory of length $T$. Each batch, therefore, consists of multiple such chunks, which can be thought of as sequences starting from different points along the trajectory. This approach effectively simulates having multiple sequences generated from different initial conditions, providing diversity in the training data.
>
> The default batch size of 512 ensures that the batch size is sufficiently large to utilize the computational resources efficiently, mitigating the limitations associated with small batch and block sizes.
>
> ***Q2. Positive Definite Matrix ${M}$***
>
> Here is how the positive definiteness and symmetry properties are ensured:
>
> ${M}$ is the square matrix being inverted in the least squares solution, so it is positive definite. Specifically, ${M} \= {A}^\\top {W} {A}$ (Section 3.2) where ${A}$ is the coefficient matrix (Section 3.1) and ${W}$ is the diagonal weight matrix (Section 3.1). The inclusion of the smoothness constraints ensures that ${A}$ has full column rank. ${W}$ is a diagonal matrix where each diagonal element is the positive squared weight of the corresponding equation.
>
> For Eq. 28 in the initial submission, ${M}^{-1}$ is symmetric because ${M}$ is symmetric.
>
> We have made our statement clearer in our revised manuscript.
>
> Once again, we thank you for your valuable feedback. Your insights have helped us improve the clarity and completeness of our paper.
>
> ***References***
>
> \[1\] Adeel Pervez, Francesco Locatello, and Stratis Gavves. Mechanistic neural networks for scientific machine learning. In Forty-first International Conference on Machine Learning, ICML 2024, Vienna, Austria, July 21-27, 2024\. OpenReview.net, 2024\. URL https://openreview.net/forum?id=pLtuwhoQh7.
> \[2\] Dingling Yao, Caroline Muller, and Francesco Locatello. Marrying causal representation learning with dynamical systems for science. Advances in Neural Information Processing Systems, 37, 2024\.
> \[3\] Ricky T. Q. Chen, Yulia Rubanova, Jesse Bettencourt, and David Duvenaud. Neural ordinary differential equations. Advances in Neural Information Processing Systems, 2018\.
> \[4\] Zongyi Li, Nikola Borislavov Kovachki, Kamyar Azizzadenesheli, Burigede Liu, Kaushik Bhattacharya, Andrew M. Stuart, and Anima Anandkumar. Fourier neural operator for parametric partial differential equations. In 9th International Conference on Learning Representations, ICLR 2021, Virtual Event, Austria, May 3-7, 2021\. OpenReview.net, 2021\. URL https://openreview.net/forum?id=c8P9NQVtmnO.
> \[5\] M. Raissi, P. Perdikaris, and G.E. Karniadakis. Physics-informed neural networks: A deep learning framework for solving forward and inverse problems involving nonlinear partial differential equations. Journal of Computational Physics, 378:686-707, 2019\. ISSN 0021-9991. doi: https://doi.org/10.1016/j.jcp.2018.10.045. URL https://www.sciencedirect.com/science/article/pii/S0021999118307125.
> \[6\] Kadierdan Kaheman, J Nathan Kutz, and Steven L Brunton. Sindy-pi: a robust algorithm for parallel implicit sparse identification of nonlinear dynamics. Proceedings of the Royal Society A, 476 (2242):20200279, 2020\.
> \[7\] Francesco Locatello, Ben Poole, Gunnar Raetsch, Bernhard Sch ¨olkopf, Olivier Bachem, and Michael Tschannen. Weakly-supervised disentanglement without compromises. In Hal Daume III and Aarti Singh (eds.), Proceedings of the 37th International Conference on Machine Learning, volume 119 of Proceedings of Machine Learning Research, pp. 6348-6359. PMLR, 13-18 Jul 2020\. URL https://proceedings.mlr.press/v119/locatello20a.html.
> \[8\] Johannes Brandstetter, Max Welling, and Daniel E. Worrall. Lie point symmetry data augmentation for neural PDE solvers. In Kamalika Chaudhuri, Stefanie Jegelka, Le Song, Csaba Szepesvari, Gang Niu, and Sivan Sabato (eds.), International Conference on Machine Learning, ICML 2022, 17-23 July 2022, Baltimore, Maryland, USA, volume 162 of Proceedings of Machine Learning Research, pp. 2241-2256. PMLR, 2022\. URL https://proceedings.mlr.press/v162/brandstetter22a.html.
> \[9\] J.R. Dormand and P.J. Prince. A family of embedded runge-kutta formulae. Journal of Computational and Applied Mathematics, 6(1):19-26, 1980\. ISSN 0377-0427. doi: https://doi.org/10.1016/0771-050X(80)90013-3. URL https://www.sciencedirect.com/science/article/pii/0771050X80900133.
> \[10\] A. C. Hindmarsh. ODEPACK, a systematized collection of ODE solvers. In R. S. Stepleman (ed.), Scientific Computing, pp. 55-64, Amsterdam, 1983\. North-Holland.

---

> > ### Comment · Reviewer_bGB5 · 2024-11-23
> >
> > I thank the authors for the range of additional experiments and results that were provided.
> >
> > I am grateful for the additional explanation on non-linear ODEs and the added appendix notes on slack variables. One of the things I particularly like about the paper is that it is self-contained and does not heavily really on the original MNN. This is excellent presentation in my opinion.
> >
> > I'm happy to see the additional experiments and comparisons outside the MNN family. I think these results should increase adoption of the method, given the limited relevance of this work -- it builds upon quite a niche method (something I see reviewer y7GX pointed out as well).
> >
> > In your response to Q1, I agree with your strategy to simulate more trajectories for the Lorentz system. Like you just did, please find a way to clearly expose the contents of a batch in the final version of the paper.
> >
> > Finally, the proofs and overall training and testing pseudocodes are much clearer now. Thank you.
> >
> > All in all, I really appreciate the additional work, and I have increased my score in other areas. That said, I kept the overall __positive__ score unchanged.

---

### Official Review · Reviewer_5t9U · 2024-10-31

**Soundness:** 4
**Presentation:** 3
**Contribution:** 4
**Rating:** 8
**Confidence:** 3

**Summary:**

The authors provide an alternative method to implement ODE solving in mechanistic neural networks (MNNs) to achieve better time and space complexity without sacrificing performance.

**Strengths:**

- straightforward and good idea, fuelled by several computational tricks
- strong complexity reduction!
- clear writing
- clear motivation
- experiments on toys to illustrate the point and on real world data to show feasibility

**Weaknesses:**

- while the description of MNNs is nice and concise, i believe a reader could benefit from a small additional description of the aspects of the network that are improved upon, like the slack variables.
- I am missing some ablations studies, for instance on the importance weights. do they do anything in practice?

**Questions:**

- I don’t fully understand what determines the sparsity level of $A$. You say 95% in typical applications, which kinds of applications yield denser $A$? IIUC, the sparsity comes from the fact that none of the Eqs 5,6,7,8 entangle $y$ that are more than a timestep apart.

### Nits:
- $V$ is undefined when introducing Eq 1, only defined later
- maybe you can write out the definition of A, y and b once for clarity


### To the other reviewers
I am not an expert in this area, so take my review with a grain of salt.

---

> ### Author Response · Authors · 2024-11-21
> **Response to Reviewer 5t9U**
>
> We thank you for your positive feedback. We address your comments and questions below.
>
> ***W1: Additional Description of the Modified Components from the Original MNN***
>
> We appreciate your suggestion to provide a brief description of the slack variables and the aspects of the original MNN that we improved upon. While our focus was on introducing the improvements in our method and we did not want to confuse the readers with the components that are not present in our new method, we agree that including a brief explanation would help readers better understand the context of our contributions. In response, **we have included a concise description of the original MNN in Appendix C of the revised manuscript.**
>
> ***W2: Ablation Studies***
>
> The importance weights $w\_{\\mathrm{gov}}$, $w\_{\\mathrm{init}}$, and $w\_{\\mathrm{smooth}}$ are optional. They are introduced for generalization purposes and to **allow flexibility in balancing different constraints**. **In all our experiments, we set these weights to 1**, which means they are effectively not used. **We find that this default setting is sufficient to achieve good results across all our tasks.** We include these weights to allow future applications to adjust if needed.
>
> The weights $s\_{t}^{r}$ for the smoothness constraints are not optional, but inherent to the formulation. Also, note that they are **already present in the original MNN** (as seen in Eqs. 10 and 11 in \[1\]).
>
> Our key insight is that the solver is inherently an approximation of the true continuous-time dynamics due to time discretization. The original MNN solver is itself an approximation, and our modifications represent alternative approximation methods that improve efficiency without sacrificing accuracy. Our experiments demonstrate that the accuracy of S-MNN is comparable to that of the original MNN, indicating that our simplifications do not adversely affect performance.
>
> Regarding ablation studies on the other changes we made, due to time constraints, we were unable to include detailed ablation studies in the current submission. However, we recognize that such studies could provide valuable insights into the contributions of each modification. If the reviewer feels it would be beneficial, we will prioritize conducting these ablation experiments and include them in the next version of our work.
>
> ***Q1: Sparsity Level of ${A}$***
>
> You are correct in your understanding that the sparsity of the matrix ${A}$ arises because none of the constraints entangle variables that are more than one timestep apart. Therefore, the sparsity level is approximately $1 \- O(1/T)$. The exact sparsity depends on the application. Specifically, as $T$ increases, the proportion of non-zero elements in ${A}$ decreases, leading to higher sparsity. Conversely, for smaller $T$, ${A}$ becomes denser. The 95% sparsity is measured from the Lorenz discovery experiment (Section 5.2) with $T \= 50$. We have clarified the sparsity level in the revised manuscript.
>
> ***Minor Points***
>
> 1\. We have added the definition of $V$ (the number of dimensions) when introducing Eq. 1\.
>
> 2\. Due to space constraints in the main text, we only provided a concise description of ${A}$, ${y}$, and ${b}$ in our initial version. Following your suggestion, **we have included detailed formulations of these matrices and vectors in Appendix A.1 in the revised manuscript**. This addition should help readers better understand our linear system.
>
> Thank you again for your positive evaluation and support.
>
> ***References***
>
> \[1\] Adeel Pervez, Francesco Locatello, and Stratis Gavves. Mechanistic neural networks for scientific machine learning. In Forty-first International Conference on Machine Learning, ICML 2024, Vienna, Austria, July 21-27, 2024\. OpenReview.net, 2024\. URL https://openreview.net/forum?id=pLtuwhoQh7.

---

> > ### Comment · Reviewer_5t9U · 2024-11-21
> >
> > thanks for the clarifications. I maintain my positive score.

---

### Official Review · Reviewer_y7GX · 2024-11-04

**Soundness:** 4
**Presentation:** 3
**Contribution:** 2
**Rating:** 6
**Confidence:** 2

**Summary:**

In their paper, the authors propose an alteration to the recently published Mechanistic Neural Networks (MNN), making the architecture less computationally demanding, while maintaining its predictive power. Their method, called Scalable MNNs (S-MNN), decreases computational cost primarily by removing certain slack variables present in the original method, as well as replacing a quadratic programing problem with a simple least squares regression, which has known analytical solution and can be more efficiently computed.

The authors begin by contextualizing their work among the scientific machine learning literature, then providing a brief overview of MNNs, an architecture aimed at modeling time-series data by learning an ODE in latent space, where the inputs are first encoded to represent the dynamics of the desired ODE. The solution of this ODE is then optionally passed through a decoder to map the solution to the desired output space. The authors then go on to describe their method, S-MNN, which simplifies certain aspects of the original MNN implementation. The main difference between the two methods is the linearization of the underlying ODE, which makes it solvable using a traditional least squares regression, with known analytical solution. The authors also propose a sparse method of computation for obtaining this solution, which is easily parallelizable using GPUs. Overall, their alterations reduce the time and memory complexity of the architecture so that it scales linearly with the sequence length $T$.

Finally, the authors briefly describe other related work in the scientific machine learning literature, and carry out experiments to empirically validate their method. The authors employ S-MNNs on 1) a simple set of ODEs, as a sanity check for the architecture; 2) the discovery of underlying parameters of a Lorenz system; 3) solving the Korteweg-De Vries (KdV) PDE from data; and 4) forecasting sea surface temperatures from the SST-V2 dataset. Overall, the experiments indicate that S-MNNs perform comparable or slightly better than traditional MNNs, but at considerably lower computational cost.

The authors conclude by summarizing their work and outlining limitations and possible future directions of work, including parallelizing their implementation over the time dimension, which currently need to be solved sequentially.

**Strengths:**

### Originality
This work innovates by proposing a more computationally efficient way of implementing Mechanistic Neural Networks (MNNs), making it more scalable for longer roll-outs and numerically stable. Although the paper largely basis itself on the recent work by Pervez et all., 2024, they improve upon this method in an original way, and present practical ways of making the MNN strategy less computationally expensive.


### Quality
The submission appears to be well thought-out, making precise contributions to the previous work of MNNs. The simplification of the underlying ODE solver implied by the architecture appears to yield similar performance at a computational cost 2-5 times lower, with greater grains when the sequence length $T$ is bigger. The author's computational improvements are substantial, and the theoretical arguments used to formulate and motivate S-MNNs are well crafted.


### Clarity
I was not originally familiar with MNNs, so there are parts of the paper that seemed unclear to me, but I think the authors overall do a decent job of introducing the important concepts and explaining how their alterations lead to computational efficiency.

**Weaknesses:**

### Contribution Is Specific To A Niche Method
Although the ideas in the paper seem interesting, my main criticism of this submission is that their contribution is very narrow, in the sense that it improves on a method that is generally not very employed in practice. This makes their work less relevant to the scientific machine learning community. While the authors carry out several well-designed experiments comparing MNNs and S-MNNs, it is unclear to me what the advantages of S-MNNs are over the other methods described in section 4 of the paper. Potentially including some of these other methods in the experiments carried out, and comparing their accuracy/computational cost would help contextualize their work and make a stronger case for their approach. I detail some suggestions for comparisons in the "Questions" section below.


### Errors Reported Using Absolute Error Instead of Relative Error (minor weakness)
Errors should ideally be reported as relative L2 error, instead of absolute error squared. Using relative L2 error helps contextualize the performance of the method for target functions of different magnitudes. For Example, in section 5.1 the harmonic damping problem has a solution that oscillates from $\approx -0.1$ to $\approx. 0.1$, while the population problem oscillates from 0 to $\approx 40$. Using relative L2 errors would also make their results more easily comparable against what is reported in other papers for comparable problems.


### Other comments that did not affect my score:
- [line 333] While it is great to see very close agreement between true and learned solutions, it would still be valuable to report exact errors for each benchmark, likely in the appendix.
- [figure 4] Since all predictions look very close to the ground truth solution when eyeballing the plots, it would be great to plot absolute error as well, which could help differentiate the performance of the two methods.

**Questions:**

I list below my two main suggestions. As previously mentioned, addressing the first point is what is most likely to change my current rating of the paper.

- [**Overall Relevance To Scientific Computing**] Although the authors do list related methods in scientific machine learning in section 4, it is not clear to me why improving on traditional MNNs is desirable over utilizing these other methods. Why choose to focus on MNNs? In order to answer this question, for example, the authors could run their experiments using some of the other methods listed in section 4, listing their accuracy and computational cost when compared to MNNs and S-MNNs. For example, some suggestions for comparisons are: 1) in section 5.1 and 5.2, they could run comparisons against Neural-ODEs and perhaps a traditional neural operator such as FNO or DeepONet; 2) in section 5.3 they could compare against a traditional MLP using both data and a Physics Informed loss (as detailed in line 300); 3) in section 5.4 the could compare against other autoregressive models including a traditional transformer or a neural operator. Doing so would help contextualize their work in a quantifiable way, and could help increase my score of their submission, even if S-MNNs do not beat other all other methods.

- [**Use of Relative L2 Errors**] As mentioned in the previous section, reporting errors using relative L2 error instead of MSE/RMSE provides a more reliable way of comparing methods across different benchmarks, since solutions can have drastically different magnitudes. Is there a particular reason why reporting absolute MSE/RMSE makes more sense for your benchmarks?

---

> ### Author Response · Authors · 2024-11-21
> **Response to Reviewer y7GX (1/2)**
>
> We thank you for your thoughtful and constructive feedback. We are pleased that you found our work original and well thought-out, and we appreciate the opportunity to clarify and strengthen our submission. We address your concerns and questions individually below.
>
> ***1\. Overall Relevance to Scientific Computing***
>
> Thank you for raising this important point. We agree that situating our work within the broader context of scientific machine learning is crucial for understanding its relevance and impact.
>
> While **MNN is a very recent development**, introduced by Pervez et al. (2024) \[1\] and published at ICML in July this year, **it is understandable that it has not yet been widely employed in practice**. However, we believe that **MNN is a promising approach worth improving because it addresses several limitations of existing methods,** such as:
>
> * Neural ODEs \[3\]: While effective for modeling continuous-time dynamics, Neural ODEs primarily focus on forecasting and do not provide explicit mechanistic insights or governing equations.
>
> * Neural Operators \[4\]: Neural Operators do not explicitly model time evolution, which can limit their ability to capture temporal dynamics in certain applications.
>
> * PINNs \[5\]: PINNs incorporate known physics but can be challenging to train and may not generalize well to unseen conditions.
>
> In contrast, MNN learns explicit internal ODE representations from observational data, enabling not only accurate forecasting but also providing interpretable models that facilitate downstream scientific analyses, such as parameter identification and causal effect estimation \[2\].
>
> **A critical limitation of the original MNN is its lack of scalability due to high computational resource requirements**, which hinders its application to long temporal sequences and large-scale systems. Our proposed S-MNN enhances the scalability of MNN by reducing computational complexity to linear with respect to sequence length. **We believe that with our improvements, MNN has the potential to become more widely applicable and useful in the scientific machine learning community, enabling researchers to apply mechanistic modeling to real-world problems that were previously intractable with MNN.**
>
> **We agree that direct comparisons with other contemporary methods would strengthen our work. We have included the following tables and figures in the revised manuscript of our submission.**
>
> * In Appendix B.1, Table 3, we provide the error and runtime for the solver validations (sanity checks) and the comparisons to the classic solvers RK45 \[9\] and LSODA \[10\]. S-MNN achieves excellent agreement with closed-form solutions. While S-MNN may not be the most efficient solver for these pure ODE-solving tasks, it offers additional features, such as batched GPU processing and differentiability, which are not available in the classical solvers.
>
> * In Appendix B.2, Table 4, we provide the discovered Lorenz coefficients and the comparisons to SINDy \[6\]. Our S-MNN closely matches SINDy, with only minor differences observed. Neural ODE \[3\] and FNO \[4\] do not do discovery tasks of this kind. This table is already in the initial submission.
>
> * In Section 5.3, Table 2: We provide the normalized KdV prediction error (NMSE) and the comparisons to ResNet and FNO \[8\]. S-MNN significantly outperforms the ResNet and FNO models.
>
> * In Section 5.4, Figure 5: We provide the prediction error and performance for the sea surface temperature prediction task and the comparisons to the Ada-GVAE \[7\] (a plain MLP decoder) baseline. S-MNN has much smaller prediction errors than Ada-GVAE.
>
> These results demonstrate that MNN, enhanced by our S-MNN improvements, can outperform other methods and become a key player in scientific simulation. Our paper removes the key obstacle of scalability, making it possible to apply MNNs to longer sequences and larger systems. We believe that with our contributions, MNNs can now be more widely adopted, similar to how Neural ODEs \[3\] and SINDy \[6\] have influenced the field.
>
> ***2\. Use of Relative L2 Errors***
>
> We apologize for any confusion regarding the error metrics used in our paper. In our experiments, **we have indeed reported errors using relative measures**:
>
> * In Section 5.1 (Standalone Validation), we calculated the mean squared error (MSE). These errors were computed **relative** to the magnitude of the solution.
>
> * In Section 5.3 (KdV Experiment), we stated that the RMSE refers to the **relative** mean squared error. This has been changed to another metric (**normalized** mean squared error, NMSE) in the revised manuscript.
>
> * In Section 5.4 (SST Forecasting), we have mentioned that the MSE is computed over **standardized** data, effectively normalizing the errors and making them relative.
>
> To make this clearer, we have explicitly restated in the table headers and figure captions that the errors are relative.

---

> ### Author Response · Authors · 2024-11-21
> **Response to Reviewer y7GX (2/2)**
>
> ***Other Comments***
>
> Thank you for this suggestion. In the revised appendix, we have included the requested tables and figures:
>
> * The error metrics for each validation benchmark in Appendix B.1, Table 3\.
>
> * The KdV prediction error visualizations in Appendix B.3, Figure 6\.
>
> Once again, we thank you for the valuable feedback. Your insights have helped us strengthen our paper and better convey our contributions.
>
> ***References***
>
> \[1\] Adeel Pervez, Francesco Locatello, and Stratis Gavves. Mechanistic neural networks for scientific machine learning. In Forty-first International Conference on Machine Learning, ICML 2024, Vienna, Austria, July 21-27, 2024\. OpenReview.net, 2024\. URL https://openreview.net/forum?id=pLtuwhoQh7.
> \[2\] Dingling Yao, Caroline Muller, and Francesco Locatello. Marrying causal representation learning with dynamical systems for science. Advances in Neural Information Processing Systems, 37, 2024\.
> \[3\] Ricky T. Q. Chen, Yulia Rubanova, Jesse Bettencourt, and David Duvenaud. Neural ordinary differential equations. Advances in Neural Information Processing Systems, 2018\.
> \[4\] Zongyi Li, Nikola Borislavov Kovachki, Kamyar Azizzadenesheli, Burigede Liu, Kaushik Bhattacharya, Andrew M. Stuart, and Anima Anandkumar. Fourier neural operator for parametric partial differential equations. In 9th International Conference on Learning Representations, ICLR 2021, Virtual Event, Austria, May 3-7, 2021\. OpenReview.net, 2021\. URL https://openreview.net/forum?id=c8P9NQVtmnO.
> \[5\] M. Raissi, P. Perdikaris, and G.E. Karniadakis. Physics-informed neural networks: A deep learning framework for solving forward and inverse problems involving nonlinear partial differential equations. Journal of Computational Physics, 378:686-707, 2019\. ISSN 0021-9991. doi: https://doi.org/10.1016/j.jcp.2018.10.045. URL https://www.sciencedirect.com/science/article/pii/S0021999118307125.
> \[6\] Kadierdan Kaheman, J Nathan Kutz, and Steven L Brunton. Sindy-pi: a robust algorithm for parallel implicit sparse identification of nonlinear dynamics. Proceedings of the Royal Society A, 476 (2242):20200279, 2020\.
> \[7\] Francesco Locatello, Ben Poole, Gunnar Raetsch, Bernhard Sch ¨olkopf, Olivier Bachem, and Michael Tschannen. Weakly-supervised disentanglement without compromises. In Hal Daume III and Aarti Singh (eds.), Proceedings of the 37th International Conference on Machine Learning, volume 119 of Proceedings of Machine Learning Research, pp. 6348-6359. PMLR, 13-18 Jul 2020\. URL https://proceedings.mlr.press/v119/locatello20a.html.
> \[8\] Johannes Brandstetter, Max Welling, and Daniel E. Worrall. Lie point symmetry data augmentation for neural PDE solvers. In Kamalika Chaudhuri, Stefanie Jegelka, Le Song, Csaba Szepesvari, Gang Niu, and Sivan Sabato (eds.), International Conference on Machine Learning, ICML 2022, 17-23 July 2022, Baltimore, Maryland, USA, volume 162 of Proceedings of Machine Learning Research, pp. 2241-2256. PMLR, 2022\. URL https://proceedings.mlr.press/v162/brandstetter22a.html.
> \[9\] J.R. Dormand and P.J. Prince. A family of embedded runge-kutta formulae. Journal of Computational and Applied Mathematics, 6(1):19-26, 1980\. ISSN 0377-0427. doi: https://doi.org/10.1016/0771-050X(80)90013-3. URL https://www.sciencedirect.com/science/article/pii/0771050X80900133.
> \[10\] A. C. Hindmarsh. ODEPACK, a systematized collection of ODE solvers. In R. S. Stepleman (ed.), Scientific Computing, pp. 55-64, Amsterdam, 1983\. North-Holland.

---

> > ### Comment · Reviewer_y7GX · 2024-11-22
> >
> > I thank the authors for their time reading the reviews and making improvements to the paper, as well as answering our questions.
> >
> > Based on their revisions and further consideration about the relevance of their work, I will be increasing the score of their submission from a 5 to a 6. While I see the useful technical contributions they propose, I still maintain that since it primarily improves upon a method that has just recently been published and not necessarily widely adopted (at least as of now), the relevance of the paper becomes narrow, preventing me from increasing the score to an 8, which is the next available score.
> >
> > I thank all the authors once again for their careful response and improvements to the paper, and hope my review has been useful.

---

### Official Review · Reviewer_1WLp · 2024-11-06

**Soundness:** 3
**Presentation:** 2
**Contribution:** 3
**Rating:** 6
**Confidence:** 2

**Summary:**

The paper presents Scalable Mechanistic Neural Network (S-MNN) designed for long temporal sequences.
S-MNN is more efficient compared to a previously proposed solution MNN as it reduces the computational time and space to linear in the sequnce length.
The paper describes the algorithmic improvements to reduce the computational cost and provide experimental results

**Strengths:**

- The problem is well-motvated
- S-MNN seems to achieve good performance while improving the computational cost
- Detailed experimental results are presented

**Weaknesses:**

As discussed by the authors in the limitation section, parallelism along the time dimension can be challenging.

**Questions:**

- Are there any cases where intuitively MNN should outperform S-MNN?

---

> ### Author Response · Authors · 2024-11-21
> **Response to Reviewer 1WLp**
>
> We thank you for the positive review\!
>
> Below is our answer to your question "*Are there any cases where intuitively MNN should outperform S-MNN?"*
>
> In tasks involving very short temporal sequences or small-scale systems, the computational overhead associated with the specialized solver in S-MNN may not be justified. The original MNN, despite its higher asymptotic computational complexities, might perform faster due to lower constant factors and simpler data structures. In such cases, the computational overhead of setting up the banded matrices and optimized solver in S-MNN could outweigh the benefits, making the original MNN more efficient.
>
> However, we would like to note that in such small-scale problems, both S-MNN and MNN use very little computational resources, so the absolute difference is tiny. **Therefore, while the original MNN might have a slight edge in efficiency for these specific small-scale cases, the practical impact is often negligible.**
>
> **In terms of accuracy, we do not find noticeable differences between S-MNN and MNN in practice.**
>
> We hope this clarifies your question.

---

### Author Response · Authors · 2024-11-21
**Major Modifications in the Updated Manuscript**

We are extremely grateful to all reviewers for taking the time to review this paper, and we thank all reviewers for their valuable feedback.

We have made several changes in the updated manuscript. The modifications are marked in red font, and we have added margin marks (FIX or NEW) for them.

We summarize the major changes below.

**Section 5.3, Table 2:** We provide the normalized KdV prediction error (NMSE) and the comparisons to ResNet and FNO \[8\]. S-MNN significantly outperforms the ResNet and FNO models.

**Section 5.4, Figure 5:** We provide the prediction error and performance for the sea surface temperature prediction task and the comparisons to the Ada-GVAE \[7\] (a plain MLP decoder), which was also used in \[2\] as a baseline. S-MNN has much smaller prediction errors than Ada-GVAE.

**Appendix A.1:** We add detailed formulation information of ${A}$, ${b}$, ${W}$, and ${y}$.

**Appendix A.4:** We add descriptions and pseudo-codes of the training and testing procedures.

**Appendix B.1, Table 3:** We provide the error and runtime for the solver validations (sanity checks) and the comparisons to the classic solvers RK45 \[9\] and LSODA \[10\]. S-MNN achieves excellent agreement with the other solvers and the closed-form solutions.

**Appendix B.3, Figure 6:** This figure was Figure 4 in the initial submission. We add the error visualizations and move this figure to the appendix.

**Appendix C:** We include an additional description of the modified components from the original MNN \[1\].

References

\[1\] Adeel Pervez, Francesco Locatello, and Stratis Gavves. Mechanistic neural networks for scientific machine learning. In Forty-first International Conference on Machine Learning, ICML 2024, Vienna, Austria, July 21-27, 2024\. OpenReview.net, 2024\. URL https://openreview.net/forum?id=pLtuwhoQh7.
\[2\] Dingling Yao, Caroline Muller, and Francesco Locatello. Marrying causal representation learning with dynamical systems for science. Advances in Neural Information Processing Systems, 37, 2024\.
\[3\] Ricky T. Q. Chen, Yulia Rubanova, Jesse Bettencourt, and David Duvenaud. Neural ordinary differential equations. Advances in Neural Information Processing Systems, 2018\.
\[4\] Zongyi Li, Nikola Borislavov Kovachki, Kamyar Azizzadenesheli, Burigede Liu, Kaushik Bhattacharya, Andrew M. Stuart, and Anima Anandkumar. Fourier neural operator for parametric partial differential equations. In 9th International Conference on Learning Representations, ICLR 2021, Virtual Event, Austria, May 3-7, 2021\. OpenReview.net, 2021\. URL https://openreview.net/forum?id=c8P9NQVtmnO.
\[5\] M. Raissi, P. Perdikaris, and G.E. Karniadakis. Physics-informed neural networks: A deep learning framework for solving forward and inverse problems involving nonlinear partial differential equations. Journal of Computational Physics, 378:686-707, 2019\. ISSN 0021-9991. doi: https://doi.org/10.1016/j.jcp.2018.10.045. URL https://www.sciencedirect.com/science/article/pii/S0021999118307125.
\[6\] Kadierdan Kaheman, J Nathan Kutz, and Steven L Brunton. Sindy-pi: a robust algorithm for parallel implicit sparse identification of nonlinear dynamics. Proceedings of the Royal Society A, 476 (2242):20200279, 2020\.
\[7\] Francesco Locatello, Ben Poole, Gunnar Raetsch, Bernhard Sch ¨olkopf, Olivier Bachem, and Michael Tschannen. Weakly-supervised disentanglement without compromises. In Hal Daume III and Aarti Singh (eds.), Proceedings of the 37th International Conference on Machine Learning, volume 119 of Proceedings of Machine Learning Research, pp. 6348-6359. PMLR, 13-18 Jul 2020\. URL https://proceedings.mlr.press/v119/locatello20a.html.
\[8\] Johannes Brandstetter, Max Welling, and Daniel E. Worrall. Lie point symmetry data augmentation for neural PDE solvers. In Kamalika Chaudhuri, Stefanie Jegelka, Le Song, Csaba Szepesvari, Gang Niu, and Sivan Sabato (eds.), International Conference on Machine Learning, ICML 2022, 17-23 July 2022, Baltimore, Maryland, USA, volume 162 of Proceedings of Machine Learning Research, pp. 2241-2256. PMLR, 2022\. URL https://proceedings.mlr.press/v162/brandstetter22a.html.
\[9\] J.R. Dormand and P.J. Prince. A family of embedded runge-kutta formulae. Journal of Computational and Applied Mathematics, 6(1):19-26, 1980\. ISSN 0377-0427. doi: https://doi.org/10.1016/0771-050X(80)90013-3. URL https://www.sciencedirect.com/science/article/pii/0771050X80900133.
\[10\] A. C. Hindmarsh. ODEPACK, a systematized collection of ODE solvers. In R. S. Stepleman (ed.), Scientific Computing, pp. 55-64, Amsterdam, 1983\. North-Holland.

---

### Meta-Review · Area_Chair_wRNz · 2024-12-17

**Metareview:**

The paper extends the "mechanistic neural network" (MNN) framework, proposed this year at ICML. The MNN encodes the dynamics of a system in an interpretable ODE, which is then solved by an ODE solver and passed to a neural decoder. The extended version proposed here has significantly better computational complexity, which extends its possible applicability.

The paper had 4 four reviews, all positive after the rebuttal. Reviewers are generally happy about the the presentation, the technical novelty, and the possible impact. All the reviews are discussed better below. The only remaining concern is the scope of the paper, which extends a very recent method which, in the words of one reviewer, "has just recently been published and [is] not necessarily widely adopted (at least as of now)".

In light of (a) the technical validity of the paper, (b) the enthusiasm of some reviewers, and (c) the importance of AI for science, I am leaning towards acceptance.

**Additional Comments On Reviewer Discussion:**

- **Reviewer bGB5** had a few questions (e.g., clarifications on the class of problems that can be addressed by the new MNN), but they were addressed during the rebuttal. The reviewer praised the exposition and the potential impact of the paper.

- **Reviewer 5t9U** was a very shot review with minor concerns. It did not impact my final evaluation.

- **Reviewer y7GX** made a very complete review, highlighting two concerns (on the choice of metrics, and on the scope of the work). The former issue was solved during rebuttal, while the latter is discussed above.

- **Reviewer 1WLp** was an extremely short review with only vague comments. The reviewer did not answer to my request for changes, and also did not interact during the rebuttal.

---

### Decision · Program_Chairs · 2025-01-22

Accept (Poster)